# Implicit Regularization in Over-Parameterized Support Vector Machine

**Yang Sui**[*]   **Xin He**[*]   **Yang Bai**[†]
School of Statistics and Management
Shanghai University of Finance and Economics
suiyang1027@stu.sufe.edu.cn;he.xin17@mail.shufe.edu.cn;
statbyang@mail.shufe.edu.cn

## Abstract

In this paper, we design a regularization-free algorithm for high-dimensional support vector machines (SVMs) by integrating over-parameterization with Nesterov's smoothing method, and provide theoretical guarantees for the induced implicit regularization phenomenon. In particular, we construct an over-parameterized hinge loss function and estimate the true parameters by leveraging regularization-free gradient descent on this loss function. The utilization of Nesterov's method enhances the computational efficiency of our algorithm, especially in terms of determining the stopping criterion and reducing computational complexity. With appropriate choices of initialization, step size, and smoothness parameter, we demonstrate that unregularized gradient descent achieves a near-oracle statistical convergence rate. Additionally, we verify our theoretical findings through a variety of numerical experiments and compare the proposed method with explicit regularization. Our results illustrate the advantages of employing implicit regularization via gradient descent in conjunction with over-parameterization in sparse SVMs.

## 1  Introduction

In machine learning, over-parameterized models, such as deep learning models, are commonly used for regression and classification [22, 36, 31, 10]. The corresponding optimization tasks are primarily tackled using gradient-based methods. Despite challenges posed by the nonconvexity of the objective function and over-parameterization [40], empirical observations show that even simple algorithms, such as gradient descent, tend to converge to a global minimum. This phenomenon is known as the implicit regularization of variants of gradient descent, which acts as a form of regularization without an explicit regularization term [30, 29].

Implicit regularization has been extensively studied in many classical statistical problems, such as linear regression [35, 23, 44] and matrix factorization [14, 24, 3]. These studies have shown that unregularized gradient descent can yield optimal estimators under certain conditions. However, a deeper understanding of implicit regularization in classification problems, particularly in support vector machines (SVMs), remains limited. Existing studies have focused on specific cases and alternative regularization approaches [33, 27, 2]. A comprehensive analysis of implicit regularization via direct gradient descent in SVMs is still lacking. We need further investigation to explore the implications and performance of implicit regularization in SVMs.

The practical significance of such exploration becomes evident when considering today's complex data landscapes and the challenges they present. In modern applications, we often face classification

---

[*]Equal contributions.
[†]Corresponding author.

37th Conference on Neural Information Processing Systems (NeurIPS 2023).

challenges due to redundant features. Data sparsity is notably evident in classification fields like finance, document classification, and gene expression analysis. For instance, in genomics, only a few genes out of thousands are used for disease diagnosis and drug discovery [17]. Similarly, spam classifiers rely on a small selection of words from extensive dictionaries [1]. These scenarios highlight limitations in standard SVMs and those with explicit regularization, like the $\ell_2$ norm. From an applied perspective, incorporating sparsity into SVMs is an intriguing research direction. While sparsity in regression has been deeply explored recently, sparse SVMs have received less attention. Discussions typically focus on generalization error and risk analysis, with limited mention of variable selection and error bounds [32]. Our study delves into the implicit regularization in classification, complementing the ongoing research in sparse SVMs.

In this paper, we focus on the implicit regularization of gradient descent applied to high-dimensional sparse SVMs. Contrary to existing studies on implicit regularization in first-order iterative methods for nonconvex optimization that primarily address regression, we investigate this intriguing phenomenon of gradient descent applied to SVMs with hinge loss. By re-parameterizing the parameters using the Hadamard product, we introduce a novel approach to nonconvex optimization problems. With proper initialization and parameter tuning, our proposed method achieves the desired statistical convergence rate. Extensive simulation results reveal that our method outperforms the estimator under explicit regularization in terms of estimation error, prediction accuracy, and variable selection. Moreover, it rivals the performance of the gold standard oracle solution, assuming the knowledge of true support.

## 1.1 Our Contributions

First, we reformulate the parameter $\boldsymbol{\beta}$ as $\mathbf{w} \odot \mathbf{w} - \mathbf{v} \odot \mathbf{v}$, where $\odot$ denotes the Hadamard product operator, resulting in a non-convex optimization problem. This re-parameterization technique has not been previously employed for classification problems. Although it introduces some theoretical complexities, it is not computationally demanding. Importantly, it allows for a detailed theoretical analysis of how signals change throughout iterations, offering a novel perspective on the dynamics of gradient descent not covered in prior works [13, 33]. With the help of re-parameterization, we provide a theoretical analysis showing that with appropriate choices of initialization size $\alpha$, step size $\eta$, and smoothness parameter $\gamma$, our method achieves a near-oracle rate of $\sqrt{s \log p / n}$, where $s$ is the number of the signals, $p$ is the dimension of $\boldsymbol{\beta}$, and $n$ is the sample size. Notice that the near-oracle rate is achievable via explicit regularization [32, 46, 20]. To the best of our knowledge, this is the first study that investigates implicit regularization via gradient descent and establishes the near-oracle rate specifically in classification.

Second, we employ a simple yet effective smoothing technique [28, 45] for the re-parameterized hinge loss, addressing the non-differentiability of the hinge loss. Additionally, our method introduces a convenient stopping criterion post-smoothing, which we discuss in detail in Section 2.2. Notably, the smoothed gradient descent algorithm is not computationally demanding, primarily involving vector multiplication. The variables introduced by the smoothing technique mostly take values of $0$ or $1$ as smoothness parameter $\gamma$ decreases, further streamlining the computation. Incorporating Nesterov's smoothing is instrumental in our theoretical derivations. Directly analyzing the gradient algorithm with re-parameterization for the non-smooth hinge loss, while computationally feasible, introduces complexities in theoretical deductions. In essence, Nesterov's smoothing proves vital both computationally and theoretically.

Third, to support our theoretical results, we present finite sample performances of our method through extensive simulations, comparing it with both the $\ell_1$-regularized estimator and the gold standard oracle solution. We demonstrate that the number of iterations $t$ in gradient descent parallels the role of the $\ell_1$ regularization parameter $\lambda$. When chosen appropriately, both can achieve the near-oracle statistical convergence rate. Further insights from our simulations illustrate that, firstly, in terms of estimation error, our method generalizes better than the $\ell_1$-regularized estimator. Secondly, for variable selection, our method significantly reduces false positive rates. Lastly, due to the efficient transferability of gradient information among machines, our method is easier to be paralleled and generalized to large-scale applications. Notably, while our theory is primarily based on the Sub-Gaussian distribution assumption, our method is actually applicable to a much wider range. Additional simulations under heavy-tailed distributions still yield remarkably desired results. Extensive experimental results indicate that our method's performance, employing implicitly regularized gradient descent in SVMs,

rivals that of algorithms using explicit regularization. In certain simple scenarios, it even matches the performance of the oracle solution.

## 1.2 Related Work

Frequent empirical evidence shows that simple algorithms such as (stochastic) gradient descent tend to find the global minimum of the loss function despite nonconvexity. To understand this phenomenon, studies by [30, 29, 19, 38, 43] proposed that generalization arises from the implicit regularization of the optimization algorithm. Specifically, these studies observe that in over-parameterized statistical models, although optimization problems may contain bad local errors, optimization algorithms, typically variants of gradient descent, exhibit a tendency to avoid these bad local minima and converge towards better solutions. Without adding any regularization term in the optimization objective, the implicit preference of the optimization algorithm itself plays the role of regularization.

Implicit regularization has attracted significant attention in well-established statistical problems, including linear regression [13, 35, 39, 23, 44] and matrix factorization [14, 24, 3, 25, 26]. In high-dimensional sparse linear regression problems, [35, 44] introduced a re-parameterization technique and demonstrated that unregularized gradient descent yields an estimator with optimal statistical accuracy under the Restricted Isometric Property (RIP) assumption [6]. [11] obtained similar results for the single index model without the RIP assumption. In low-rank matrix sensing, [14, 24] revealed that gradient descent biases towards the minimum nuclear norm solution when initiated close to the origin. Additionally, [3] demonstrated the same implicit bias towards the nuclear norm using a depth-$N$ linear network. Nevertheless, research on the implicit regularization of gradient descent in classification problems remains limited. [33] found that the gradient descent estimator converges to the direction of the max-margin solution on unregularized logistic regression problems. In terms of the hinge loss in SVMs, [2] provided a diagonal descent approach and established its regularization properties. However, these investigations rely on the diagonal regularization process [4], and their algorithms' convergence rates depend on the number of iterations, and are not directly compared with those of explicitly regularized algorithms. Besides, Frank-Wolfe method and its variants have been used for classification [21, 34]. However, sub-linear convergence to the optimum requires the strict assumption that both the direction-finding step and line search step are performed exactly [18].

## 2 Model and Algorithms

### 2.1 Notations

Throughout this work, we denote vectors with boldface letters and real numbers with normal font. Thus, $\mathbf{w}$ denotes a vector and $w_i$ denotes the $i$-th coordinate of $\mathbf{w}$. We use $[n]$ to denote the set $\{1, 2 \ldots, n\}$. For any subset $S$ in $[n]$ and a vector $\mathbf{w}$, we use $\mathbf{w}_S$ to denote the vector whose $i$-th entry is $w_i$ if $i \in S$ and 0 otherwise. For any given vector $\mathbf{w}$, let $\|\mathbf{w}\|_1$, $\|\mathbf{w}\|$ and $\|\mathbf{w}\|_\infty$ denote its $\ell_1$, $\ell_2$ and $\ell_\infty$ norms. Moreover, for any two vectors $\mathbf{w}, \mathbf{v} \in \mathbb{R}^p$, we define $\mathbf{w} \odot \mathbf{v} \in \mathbb{R}^p$ as the Hadamard product of vectors $\mathbf{w}$ and $\mathbf{v}$, whose components are $w_i v_i$ for $i \in [p]$. For any given matrix $\mathbf{X} \in \mathbb{R}^{p_1 \times p_2}$, we use $\|\mathbf{X}\|_F$ and $\|\mathbf{X}\|_S$ to represent the Frobenius norm and spectral norm of matrix $\mathbf{X}$, respectively. In addition, we let $\{a_n, b_n\}_{n \geq 1}$ be any two positive sequences. We write $a_n \lesssim b_n$ if there exists a universal constant $C$ such that $a_n \leq C \cdot b_n$ and we write $a_n \asymp b_n$ if we have $a_n \lesssim b_n$ and $b_n \lesssim a_n$. Moreover, $a_n = \mathcal{O}(b_n)$ shares the same meaning with $a_n \lesssim b_n$.

### 2.2 Over-parameterization for $\ell_1$-regularized SVM

Given a random sample $\mathcal{Z}^n = \{(\mathbf{x}_i, y_i)\}_{i=1}^n$ with $\mathbf{x}_i \in \mathbb{R}^p$ denoting the covariates and $y_i \in \{0, 1\}$ denoting the corresponding label, we consider the following $\ell_1$-regularized SVM:

$$\min_{\boldsymbol{\beta} \in \mathbb{R}^p} \frac{1}{n} \sum_{i=1}^n (1 - y_i \mathbf{x}_i^T \boldsymbol{\beta})_+ + \lambda \|\boldsymbol{\beta}\|_1, \tag{1}$$

where $(\cdot)_+ = \max\{\cdot, 0\}$ denotes the hinge loss and $\lambda$ denotes the $\ell_1$ regularization parameter. Let $L_{\mathcal{Z}^n}(\boldsymbol{\beta})$ denote the first term of the right-hand side of (1). Previous works have shown that the $\ell_1$-regularized estimator of the optimization problem (1) and its extensions achieve a near-oracle rate of convergence to the true parameter $\boldsymbol{\beta}^*$ [32, 37, 43]. In contrast, rather than imposing the $\ell_1$

regularization term in (1), we minimize the hinge loss function $L_{\mathcal{Z}^n}(\boldsymbol{\beta})$ directly to obtain a sparse estimator. Specifically, we re-parameterize $\boldsymbol{\beta}$ as $\boldsymbol{\beta} = \mathbf{w} \odot \mathbf{w} - \mathbf{v} \odot \mathbf{v}$, using two vectors $\mathbf{w}$ and $\mathbf{v}$ in $\mathbb{R}^p$. Consequently, $L_{\mathcal{Z}^n}(\boldsymbol{\beta})$ can be reformulated as $\mathcal{E}_{\mathcal{Z}^n}(\mathbf{w}, \mathbf{v})$:

$$\mathcal{E}_{\mathcal{Z}^n}(\mathbf{w}, \mathbf{v}) = \frac{1}{n} \sum_{i=1}^{n} \left(1 - y_i \mathbf{x}_i^T (\mathbf{w} \odot \mathbf{w} - \mathbf{v} \odot \mathbf{v})\right)_+. \tag{2}$$

Note that the dimensionality of $\boldsymbol{\beta}$ in (1) is $p$, but a $2p$-dimensional parameter is involved in (2). This indicates that we over-parameterize $\boldsymbol{\beta}$ via $\boldsymbol{\beta} = \mathbf{w} \odot \mathbf{w} - \mathbf{v} \odot \mathbf{v}$ in (2). We briefly describe our motivation on over-parameterizing $\boldsymbol{\beta}$ this way. Following [16], $\|\boldsymbol{\beta}\|_1 = \mathrm{argmin}_{\boldsymbol{\beta}=\mathbf{a}\odot\mathbf{c}}(\|\mathbf{a}\|^2 + \|\mathbf{c}\|^2)/2$. Thus, the optimization problem (1) translates to $\min_{\mathbf{a},\mathbf{c}} \mathcal{E}_{\mathcal{Z}^n}(\mathbf{a}, \mathbf{c}) + \lambda(\|\mathbf{a}\|^2 + \|\mathbf{c}\|^2)/2$. For a better understanding of implicit regularization by over-parameterization, we set $\mathbf{w} = \frac{\mathbf{a}+\mathbf{c}}{2}$ and $\mathbf{v} = \frac{\mathbf{a}-\mathbf{c}}{2}$, leading to $\boldsymbol{\beta} = \mathbf{w} \odot \mathbf{w} - \mathbf{v} \odot \mathbf{v}$. This incorporation of new parameters $\mathbf{w}$ and $\mathbf{v}$ effectively over-parameterizes the problem. Finally, we drop the explicit $\ell_2$ regularization term $\lambda(\|\mathbf{a}\|^2 + \|\mathbf{c}\|^2)/2$ and perform gradient descent to minimize the empirical loss $\mathcal{E}_{\mathcal{Z}^n}(\mathbf{w}, \mathbf{v})$ in (2), in line with techniques seen in neural network training, high-dimensional regression [35, 23, 44], and high-dimensional single index models [11].

## 2.3 Nesterov's smoothing

It is well-known that the hinge loss function is not differentiable. As a result, traditional first-order optimization methods, such as the sub-gradient and stochastic gradient methods, converge slowly and are not suitable for large-scale problems [45]. Second-order methods, like the Newton and Quasi-Newton methods, can address this by replacing the hinge loss with a differentiable approximation [8]. Although these second-order methods might achieve better convergence rates, the computational cost associated with computing the Hessian matrix in each iteration is prohibitively high. Clearly, optimizing (2) using gradient-based methods may not be the best choice.

To address the trade-off between convergence rate and computational cost, we incorporate Nesterov's method [28] to smooth the hinge loss and then update the parameters via gradient descent. By employing Nesterov's method, (2) can be reformulated as the following saddle point function:

$$\mathcal{E}_{\mathcal{Z}^n}(\mathbf{w}, \mathbf{v}) \equiv \max_{\boldsymbol{\mu} \in \mathcal{P}_1} \frac{1}{n} \sum_{i=1}^{n} \left(1 - y_i \mathbf{x}_i^T (\mathbf{w} \odot \mathbf{w} - \mathbf{v} \odot \mathbf{v})\right) \mu_i,$$

where $\mathcal{P}_1 = \{\boldsymbol{\mu} \in \mathbb{R}^n : 0 \leq \mu_i \leq 1\}$. According to [28], the above saddle point function can be smoothed by subtracting a prox-function $d_\gamma(\boldsymbol{\mu})$, where $d_\gamma(\boldsymbol{\mu})$ is a strongly convex function of $\boldsymbol{\mu}$ with a smoothness parameter $\gamma > 0$. Throughout this paper, we select the prox-function as $d_\gamma(\boldsymbol{\mu}) = \frac{\gamma}{2} \|\boldsymbol{\mu}\|^2$. Consequently, $\mathcal{E}_{\mathcal{Z}^n}(\mathbf{w}, \mathbf{v})$ can be approximated by

$$\mathcal{E}_{\mathcal{Z}^n,\gamma}^*(\mathbf{w}, \mathbf{v}) \equiv \max_{\boldsymbol{\mu} \in \mathcal{P}_1} \left\{ \frac{1}{n} \sum_{i=1}^{n} \left(1 - y_i \mathbf{x}_i^T (\mathbf{w} \odot \mathbf{w} - \mathbf{v} \odot \mathbf{v})\right) \mu_i - d_\gamma(\boldsymbol{\mu}) \right\}. \tag{3}$$

Since $d_\gamma(\boldsymbol{\mu})$ is strongly convex, $\mu_i$ can be obtained by setting the gradient of the objective function in (3) to zero and has the explicit form:

$$\mu_i = \mathrm{median} \left(0, \frac{1 - y_i \mathbf{x}_i^T (\mathbf{w} \odot \mathbf{w} - \mathbf{v} \odot \mathbf{v})}{\gamma n}, 1\right). \tag{4}$$

For each sample point $\mathcal{Z}^i = \{(\mathbf{x}_i, y_i)\}$, $i \in [n]$, we use $\mathcal{E}_{\mathcal{Z}^i}(\mathbf{w}, \mathbf{v})$ and $\mathcal{E}_{\mathcal{Z}^i,\gamma}^*(\mathbf{w}, \mathbf{v})$ to denote its hinge loss and the corresponding smoothed approximation, respectively. With different choices of $\mu_i$ for any $i \in [n]$ in (4), the explicit form of $\mathcal{E}_{\mathcal{Z}^i,\gamma}^*(\mathbf{w}, \mathbf{v})$ can be written as

$$\mathcal{E}_{\mathcal{Z}^i,\gamma}^*(\mathbf{w}, \mathbf{v}) = \begin{cases} 0, & \text{if } y_i \mathbf{x}_i^T (\mathbf{w} \odot \mathbf{w} - \mathbf{v} \odot \mathbf{v}) > 1, \\ (1 - y_i \mathbf{x}_i^T (\mathbf{w} \odot \mathbf{w} - \mathbf{v} \odot \mathbf{v}))/n - \gamma/2, & \text{if } y_i \mathbf{x}_i^T (\mathbf{w} \odot \mathbf{w} - \mathbf{v} \odot \mathbf{v}) < 1 - \gamma n, \\ (1 - y_i \mathbf{x}_i^T (\mathbf{w} \odot \mathbf{w} - \mathbf{v} \odot \mathbf{v}))^2/(2\gamma n^2), & \text{otherwise.} \end{cases}$$
$$\tag{5}$$

Note that the explicit solution (5) indicates that a larger $\gamma$ yields a smoother $\mathcal{E}_{\mathcal{Z}^i,\gamma}^*(\mathbf{w}, \mathbf{v})$ with larger approximation error, and can be considered as a parameter that controls the trade-off between smoothness and approximation accuracy. The following theorem provides the theoretical guarantee of the approximation error. The proof can be directly derived from (5), and thus is omitted here.

**Theorem 1.** *For any random sample $\mathcal{Z}^i = \{(\mathbf{x}_i, y_i)\}$, $i \in [n]$, the corresponding hinge loss $\mathcal{E}_{\mathcal{Z}^i}(\mathbf{w}, \mathbf{v})$ is bounded by its smooth approximation $\mathcal{E}^*_{\mathcal{Z}^i, \gamma}(\mathbf{w}, \mathbf{v})$, and the approximation error is completely controlled by the smooth parameter $\gamma$. For any $(\mathbf{w}, \mathbf{v})$, we have*

$$\mathcal{E}^*_{\mathcal{Z}^i, \gamma}(\mathbf{w}, \mathbf{v}) \leq \mathcal{E}_{\mathcal{Z}^i}(\mathbf{w}, \mathbf{v}) \leq \mathcal{E}^*_{\mathcal{Z}^i, \gamma}(\mathbf{w}, \mathbf{v}) + \frac{\gamma}{2}.$$

### 2.4 Implicit regularization via gradient descent

In this section, we apply gradient descent algorithm to $\mathcal{E}_{\mathcal{Z}^n, \gamma}(\mathbf{w}, \mathbf{v})$ in (3) by updating $\mathbf{w}$ and $\mathbf{v}$ to obtain the estimator of $\boldsymbol{\beta}^*$. Specifically, the gradients of (3) with respect to $\mathbf{w}$ and $\mathbf{v}$ can be directly obtained, with the form of $-2/n \sum_{i=1}^n y_i \mu_i \mathbf{x}_i \odot \mathbf{w}$ and $2/n \sum_{i=1}^n y_i \mu_i \mathbf{x}_i \odot \mathbf{v}$, respectively. Thus, the updates for $\mathbf{w}$ and $\mathbf{v}$ are given as

$$\mathbf{w}_{t+1} = \mathbf{w}_t + 2\eta \frac{1}{n} \sum_{i=1}^n y_i \mu_{t,i} \mathbf{x}_i \odot \mathbf{w}_t \quad and \quad \mathbf{v}_{t+1} = \mathbf{v}_t - 2\eta \frac{1}{n} \sum_{i=1}^n y_i \mu_{t,i} \mathbf{x}_i \odot \mathbf{v}_t, \quad (6)$$

where $\eta$ denotes the step size. Once $(\mathbf{w}_{t+1}, \mathbf{v}_{t+1})$ is obtained, we can update $\boldsymbol{\beta}$ as $\boldsymbol{\beta}_{t+1} = \mathbf{w}_{t+1} \odot \mathbf{w}_{t+1} - \mathbf{v}_{t+1} \odot \mathbf{v}_{t+1}$ via the over-parameterization of $\boldsymbol{\beta}$. Note that we cannot initialize the values of $\mathbf{w}_0$ and $\mathbf{v}_0$ as zero vectors because these vectors are stationary points of the algorithm. Given the sparsity of the true parameter $\boldsymbol{\beta}^*$ with the support $S$, ideally, $\mathbf{w}$ and $\mathbf{v}$ should be initialized with the same sparsity pattern as $\boldsymbol{\beta}^*$, with $\mathbf{w}_S$ and $\mathbf{v}_S$ being non-zero and the values outside the support $S$ being zero. However, such initialization is infeasible as $S$ is unknown. As an alternative, we initialize $\mathbf{w}_0$ and $\mathbf{v}_0$ as $\mathbf{w}_0 = \mathbf{v}_0 = \alpha \mathbf{1}_{p \times 1}$, where $\alpha > 0$ is a small constant. This initialization approach strikes a balance: it aligns with the sparsity assumption by keeping the zero component close to zero, while ensuring that the non-zero component begins with a non-zero value [11].

We summarize the details of the proposed gradient descent method for high-dimensional sparse SVM in the following Algorithm 1.

---

**Algorithm 1**: Gradient Descent Algorithm for High-Dimensional Sparse SVM.

---

**Given**: Training set $\mathcal{Z}^n$, initial value $\alpha$, step size $\eta$, smoothness parameter $\gamma$, maximum iteration number $T_1$, validation set $\widetilde{\mathcal{Z}}^n$.
**Initialize**: $\mathbf{w}_0 = \alpha \mathbf{1}_{p \times 1}$, $\mathbf{v}_0 = \alpha_1 p \times 1$, and set iteration index $t = 0$.
**While** $t < T_1$, **do**

$$\mathbf{w}_{t+1} \quad = \mathbf{w}_t + 2\eta \frac{1}{n} \sum_{i=1}^n y_i \mu_{t,i} \mathbf{x}_i \odot \mathbf{w}_t;$$

$$\mathbf{v}_{t+1} \quad = \mathbf{v}_t - 2\eta \frac{1}{n} \sum_{i=1}^n y_i \mu_{t,i} \mathbf{x}_i \odot \mathbf{v}_t;$$

$$\boldsymbol{\beta}_{t+1} \quad = \mathbf{w}_{t+1} \odot \mathbf{w}_{t+1} - \mathbf{v}_{t+1} \odot \mathbf{v}_{t+1};$$

$$\mu_{t+1,i} \quad = \text{median}\left(0, \frac{1 - y_i \mathbf{x}_i^T \boldsymbol{\beta}_{t+1}}{n\gamma}, 1\right);$$

$$t \quad = t + 1.$$

**End if** $t > T_1$ or $\boldsymbol{\mu}_t = \mathbf{0}$.
**Return** Set $\widehat{\boldsymbol{\beta}}$ as $\boldsymbol{\beta}_t$.

---

We highlight three key advantages of Algorithm 1. First, the stopping condition for Algorithm 1 can be determined based on the value of $\boldsymbol{\mu}$ in addition to the preset maximum iteration number $T_1$. Specifically, when the values of $\mu_i$ are 0 across all samples, the algorithm naturally stops as no further updates are made. Thus, $\boldsymbol{\mu}$ serves as an intrinsic indicator for convergence, providing a more efficient stopping condition. Second, Algorithm 1 avoids heavy computational cost like the computation of the Hessian matrix. The main computational load comes from the vector multiplication in (6). Since a considerable portion of the elements in $\boldsymbol{\mu}$ are either 0 or 1, and the proportion of these elements increases substantially as $\gamma$ decreases, the computation in (6) can be further simplified. Lastly, the utilization of Nesterov's smoothing not only optimizes our approach but also aids in our theoretical derivations, as detailed in Appendix E.

# 3 Theoretical Analysis

In this section, we analyze the theoretical properties of Algorithm 1. The main result is the error bound $\|\boldsymbol{\beta}_t - \boldsymbol{\beta}^*\|$, where $\boldsymbol{\beta}^*$ is the minimizer of the population hinge loss function for $\boldsymbol{\beta}$ without the $\ell_1$ norm: $\boldsymbol{\beta}^* = \arg\min_{\boldsymbol{\beta} \in \mathbb{R}^p} \mathbb{E}(1 - y\mathbf{x}^T\boldsymbol{\beta})_+$. We start by defining the $\delta$-incoherence, a key assumption for our analysis.

**Definition 1.** *Let $\mathbf{X} \in \mathbb{R}^{n \times p}$ be a matrix with $\ell_2$-normalized columns $\mathbf{x}_1, \ldots, \mathbf{x}_p$, i.e., $\|\mathbf{x}_i\| = 1$ for all $i \in [n]$. The coherence $\delta = \delta(\mathbf{X})$ of the matrix $\mathbf{X}$ is defined as*

$$\delta := \max_{K \subseteq [n], 1 \leq i \neq j \leq p} |\langle \mathbf{x}_i \odot \mathbf{1}_K, \mathbf{x}_j \odot \mathbf{1}_K \rangle|,$$

*where $\mathbf{1}_K$ denotes the $n$-dimensional vector whose $i$-th entry is $1$ if $i \in K$ and $0$ otherwise. Then, the matrix $\mathbf{X}$ is said to be satisfying $\delta$-incoherence.*

Coherence measures the suitability of measurement matrices in compressed sensing [12]. Several techniques exist for constructing matrices with low coherence. One such approach involves utilizing sub-Gaussian matrices that satisfy the low-incoherence property with high probability [9, 7]. The Restricted Isometry Property (RIP) is another key measure for ensuring reliable sparse recovery in various applications [35, 44], but verifying RIP for a designed matrix is NP-hard, making it computationally challenging [5]. In contrast, coherence offers a computationally feasible metric for sparse regression [9, 23]. Hence, the assumptions required in our main theorems can be verified within polynomial time, distinguishing them from the RIP assumption.

Recall that $\boldsymbol{\beta}^* \in \mathbb{R}^p$ is the $s$-sparse signal to be recovered. Let $S \subset \{1, \ldots, p\}$ denote the index set that corresponds to the nonzero components of $\boldsymbol{\beta}^*$, and the size $|S|$ of $S$ is given by $s$. Among the $s$ nonzero signal components of $\boldsymbol{\beta}^*$, we define the index set of strong signals as $S_1 = \{i \in S : |\beta_i^*| \geq C_s \log p \sqrt{\log p/n}\}$ and of weak signals as $S_2 = \{i \in S : |\beta_i^*| \leq C_w \sqrt{\log p/n}\}$ for some constants $C_s, C_w > 0$. We denote $s_1$ and $s_2$ as the cardinalities of $S_1$ and $S_2$, respectively. Furthermore, we use $m = \min_{i \in S_1} |\beta_i^*|$ to represent the minimal strength for strong signals and $\kappa$ to represent the condition number-the ratio of the largest absolute value of strong signals to the smallest. In this paper, we focus on the case that each nonzero signal in $\boldsymbol{\beta}^*$ is either strong or weak, which means that $s = s_1 + s_2$. Regarding the input data and parameters in Algorithm 1, we introduce the following two structural assumptions.

**Assumption 1.** *The design matrix $\mathbf{X}/\sqrt{n}$ satisfies $\delta$-incoherence with $0 < \delta \lesssim 1/(\kappa s \log p)$. In addition, every entry $x$ of $\mathbf{X}$ is i.i.d. zero-mean sub-Gaussian random variable with bounded sub-Gaussian norm $\sigma$.*

**Assumption 2.** *The initialization for gradient descent are $\mathbf{w}_0 = \alpha \mathbf{1}_{p \times 1}$, $\mathbf{v}_0 = \alpha \mathbf{1}_{p \times 1}$ where the initialization size $\alpha$ satisfies $0 < \alpha \lesssim 1/p$, the parameter of prox-function $\gamma$ satisfies $0 < \gamma \leq 1/n$, and the step size $\eta$ satisfies $0 < \eta \lesssim 1/(\kappa \log p)$.*

Assumption 1 characterizes the distribution of the input data, which can be easily satisfied across a wide range of distributions. Interestingly, although our proof relies on Assumption 1, numerical results provide compelling evidence that it isn't essential for the success of our method. This indicates that the constraints set by Assumption 1 can be relaxed in practical applications, as discussed in Section 4. The assumptions about the initialization size $\alpha$, the smoothness parameter $\gamma$, and the step size $\eta$ primarily stem from the theoretical induction of Algorithm 1. For instance, $\alpha$ controls the strength of the estimated weak signals and error components, $\gamma$ manages the approximation error in smoothing, and $\eta$ affects the accuracy of the estimation of strong signals. Our numerical simulations indicate that extremely small initialization size $\alpha$, step size $\eta$, and smoothness parameter $\gamma$ are not required to achieve the desired convergence results, highlighting the low computational burden of our method, with details found in Section 4. The primary theoretical result is summarized in the subsequent theorem.

**Theorem 2.** *Suppose that Assumptions 1 and 2 hold, then there exist positive constants $c_1, c_2, c_3$ and $c_4$ such that there holds with probability at least $1 - c_1 n^{-1} - c_2 p^{-1}$ that, for every time $t$ with $c_3 \log(m/\alpha^2)/(\eta m) \leq t \leq c_4 \log(1/\alpha)/(\eta \log n)$, the solution of the $t$-th iteration in Algorithm 1, $\boldsymbol{\beta}_t = \mathbf{w}_t \odot \mathbf{w}_t - \mathbf{v}_t \odot \mathbf{v}_t$, satisfies*

$$\|\boldsymbol{\beta}_t - \boldsymbol{\beta}^*\|^2 \lesssim \frac{s \log p}{n}.$$

Theorem 2 demonstrates that if $\boldsymbol{\beta}^*$ contains $s$ nonzero signals, then with high probability, for any $t \in [c_3 \log(m/\alpha^2)/(\eta m), c_4 \log(1/\alpha)/(\eta \log n)]$, the convergence rate of $\mathcal{O}(\sqrt{s \log p/n})$ in terms of the $\ell_2$ norm can be achieved. Such a convergence rate matches the near-oracle rate of sparse SVMs and can be attained through explicit regularization using the $\ell_1$ norm penalty [32, 46], as well as through concave penalties [20]. Therefore, Theorem 2 indicates that with over-parameterization, the implicit regularization of gradient descent achieves the same effect as imposing an explicit regularization into the objective function in (1).

**Proof Sketch.** The ideas behind the proof of Theorem 2 are as follows. First, we can control the estimated strengths associated with the non-signal and weak signal components, denoted as $\|\mathbf{w}_t \odot \mathbf{1}_{S_1^c}\|_\infty$ and $\|\mathbf{v}_t \odot \mathbf{1}_{S_1^c}\|_\infty$, to the order of the square root of the initialization size $\alpha$ for up to $\mathcal{O}(\log(1/\alpha)/(\eta \log n))$ steps. This provides an upper boundary on the stopping time. Also, the magnitude of $\alpha$ determines the size of coordinates outside the signal support $S_1$ at the stopping time. The importance of choosing small initialization sizes and their role in achieving the desired statistical performance are further discussed in Section 4. On the other hand, each entry of the strong signal part, denoted as $\boldsymbol{\beta}_t \odot \mathbf{1}_{S_1}$, increases exponentially with an accuracy of around $\mathcal{O}(\log p/n)$ near the true parameter $\boldsymbol{\beta}^* \odot \mathbf{1}_{S_1}$ within roughly $\mathcal{O}(\log(m/\alpha^2)/(\eta m))$ steps. This establishes the left boundary of the stopping time. The following two Propositions summarize these results.

**Proposition 1.** *(Analyzing Weak Signals and Errors) Under Assumptions* 1-2, *with probability at least* $1 - cn^{-1}$, *we have*

$$\|\mathbf{w}_t \odot \mathbf{1}_{S_1^c}\|_\infty \leq \sqrt{\alpha} \lesssim \frac{1}{\sqrt{p}} \quad and \quad \|\mathbf{v}_t \odot \mathbf{1}_{S_1^c}\|_\infty \leq \sqrt{\alpha} \lesssim \frac{1}{\sqrt{p}},$$

*for all* $t \leq T^* = \mathcal{O}(\log(1/\alpha)/(\eta \log n))$, *where* $c$ *is some positive constant.*

**Proposition 2.** *(Analyzing Strong Signals) Under Assumptions* 1-2, *with probability at least* $1 - c_1 n^{-1} - c_2 p^{-1}$, *we have*

$$\|\boldsymbol{\beta}_t \odot \mathbf{1}_{S_1} - \boldsymbol{\beta}^* \odot \mathbf{1}_{S_1}\|_\infty \lesssim \sqrt{\frac{\log p}{n}},$$

*holds for all* $\mathcal{O}(\log(m/\alpha^2)/(\eta m)) \leq t \leq \mathcal{O}(\log(1/\alpha)/(\eta \log n))$ *where* $c_1, c_2$ *are two constants.*

Consequently, by appropriately selecting the stopping time $t$ within the interval specified in Theorem 2, we can ensure convergence of the signal components and effectively control the error components. The final convergence rate can be obtained by combining the results from Proposition 1 and Proposition 2.

## 4    Numerical Study

In our simulations, unless otherwise specified, we follow a default setup. We generate $3n$ independent observations, divided equally for training, validation, and testing. The true parameters $\boldsymbol{\beta}^*$ is set to $m\mathbf{1}_S$ with a constant $m$. Each entry of $\mathbf{x}$ is sampled as $i.i.d.$ zero-mean Gaussian random variable, and the labels $y$ are determined by a binomial distribution with probability $p = 1/(1 + \exp(\mathbf{x}^T\boldsymbol{\beta}^*))$. Default parameters are: true signal strength $m = 10$, number of signals $s = 4$, sample size $n = 200$, dimension $p = 400$, step size $\eta = 0.5$, smoothness parameter $\gamma = 10^{-4}$, and initialization size $\alpha = 10^{-8}$. For evaluation, we measure the estimation error using $\|\boldsymbol{\beta}_t/\|\boldsymbol{\beta}_t\| - \boldsymbol{\beta}^*/\|\boldsymbol{\beta}^*\|\|$ (for comparison with oracle estimator) and the prediction accuracy on the testing set with $P(\hat{y} = y_{test})$. Additionally, "False positive" and "True negative" metrics represent variable selection errors. Specifically, "False positive" means the true value is zero but detected as a signal, while "True negative" signifies a non-zero true value that isn't detected. Results are primarily visualized employing shaded plots and boxplots, where the solid line depicts the median of 30 runs and the shaded area marks the 25-th and 75-th percentiles over these runs.

**Effects of Small Initialization Size.** We investigate the power of small initialization size $\alpha$ on the performance of our algorithm. We set the initialization size $\alpha = \{10^{-4}, 10^{-6}, 10^{-8}, 10^{-10}\}$, and other parameters are set by default. Figure 1 shows the importance of small initialization size in inducing exponential paths for the coordinates. Our simulations reveal that small initialization size leads to lower estimation errors and more precise signal recovery, while effectively constraining the error term to a negligible magnitude. Remarkably, although small initialization size might slow the convergence rate slightly, this trade-off is acceptable given the enhanced estimation accuracy.

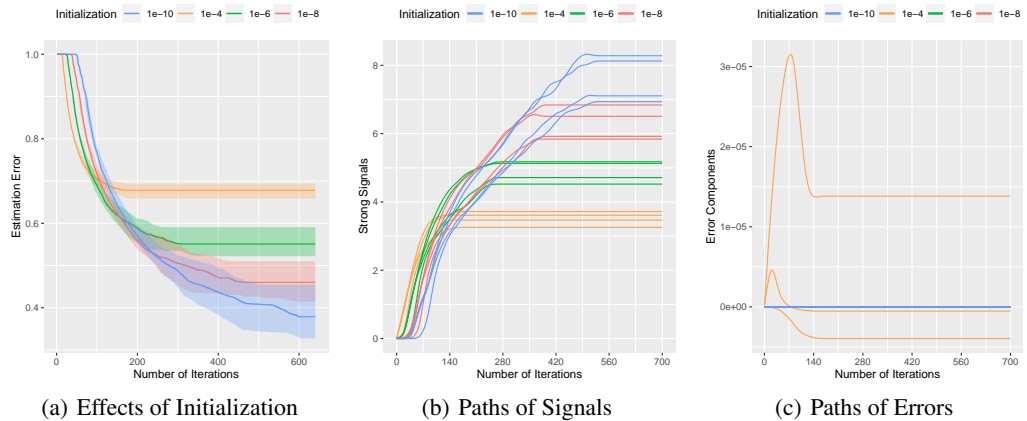

(a) Effects of Initialization     (b) Paths of Signals     (c) Paths of Errors

Figure 1: Effects of small initialization size $\alpha$. In Figure 1(a), the estimation error is calculated by $\|\boldsymbol{\beta}_t - \boldsymbol{\beta}^*\|/\|\boldsymbol{\beta}^*\|$.

**Effects of Signal Strength and Sample Size.** We examine the influence of signal strength on the estimation accuracy of our algorithm. We set the true signal strength $m = 0.5 * k, k = 1, \ldots, 20$ and keep other parameters at their default values. As depicted in Figure 2, we compare our method (denoted by **GD**) with $\ell_1$-regularized SVM (denoted by **Lasso** method), and obtain the oracle solution (denoted by **Oracle**) using the true support information. We assess the advantages of our algorithm from three aspects. Firstly, in terms of estimation error, our method consistently outperforms the Lasso method across different signal strengths, approaching near-oracle performance. Secondly, all three methods achieve high prediction accuracy on the testing set. Lastly, when comparing variable selection performance, our method significantly surpasses the Lasso method in terms of false positive error. Since the true negative error of both methods is basically 0 , we only present results for false positive error in Figure 2.

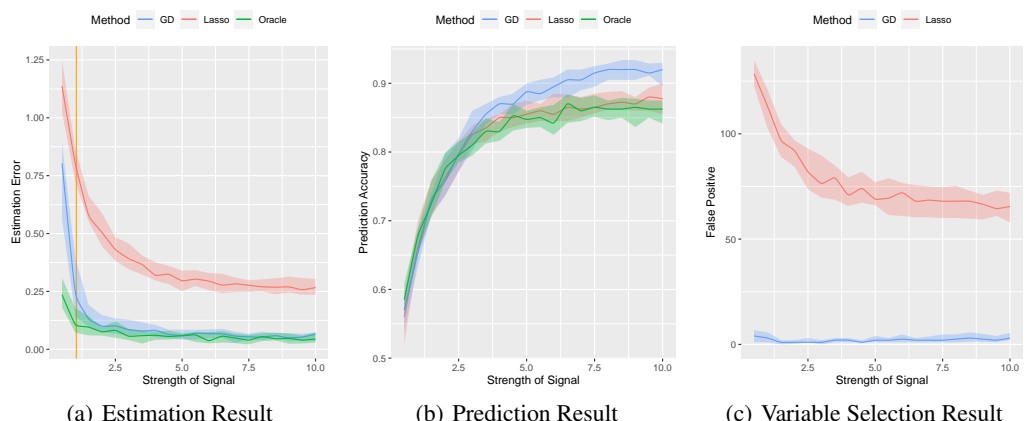

(a) Estimation Result     (b) Prediction Result     (c) Variable Selection Result

Figure 2: Effects of signal strength $m$. The orange vertical line in Figure 2(a) show the threshold of strong signal $\log p \sqrt{\log p / n}$.

We further analyze the impact of sample size $n$ on our proposed algorithm. Keeping the true signal strength fixed at $m = 5$, we vary the sample size as $n = 50 * k$ for $k = 1, \ldots, 8$. Other parameters remain at their default values. Consistently, our method outperforms the Lasso method in estimation, prediction, and variable selection, see Figure 3 for a summary of the results.

**Performance on Complex Signal Structure.** To examine the performance of our method under more complex signal structures, we select five signal structures: $\mathbf{A} - (5, 6, 7, 8)$, $\mathbf{B} - (4, 6, 8, 9)$, $\mathbf{C} - (3, 6, 9, 10)$, $\mathbf{D} - (2, 6, 10, 11)$, and $\mathbf{E} - (1, 6, 11, 12)$. Other parameters are set by default. The

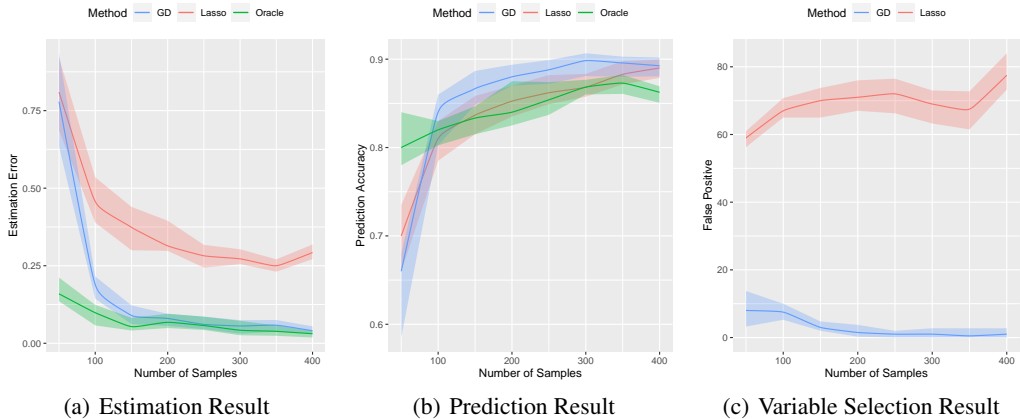

(a) Estimation Result      (b) Prediction Result      (c) Variable Selection Result

Figure 3: Effects of sample size $n$.

results, summarized in Figure 4, highlight the consistent superiority of our method over the Lasso method in terms of prediction and variable selection performance, even approaching an oracle-like performance for complex signal structures. High prediction accuracy is achieved by the both methods.

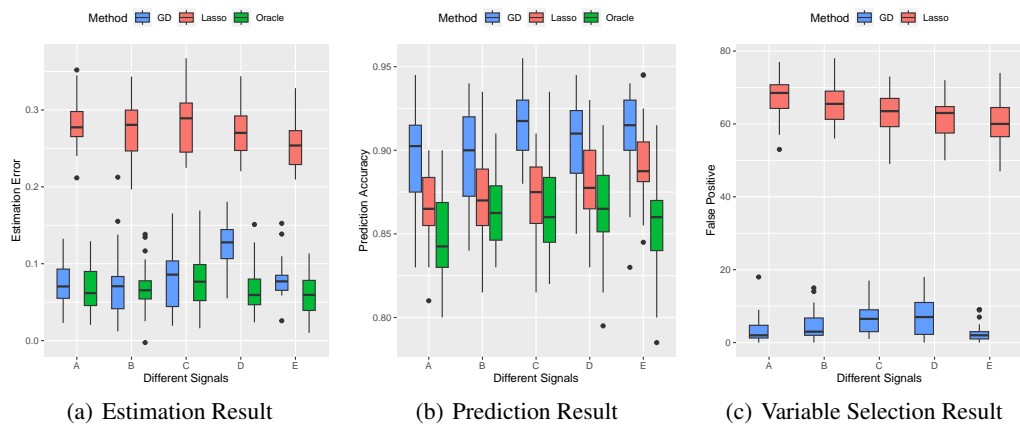

(a) Estimation Result      (b) Prediction Result      (c) Variable Selection Result

Figure 4: Performance on complex signal structure. The boxplots are depicted based on 30 runs.

**Performance on Heavy-tailed Distribution.** Although the sub-Gaussian distribution of input data is assumed in Assumption 1, we demonstrate that our method can be extended to a wider range of distributions. We conduct simulations under both uniform and heavy-tailed distributions. The simulation setup mirrors that of Figure 4, with the exception that we sample $\mathbf{x}$ from a $[-1, 1]$ uniform distribution and a $t(3)$ distribution, respectively. Results corresponding to the $t(3)$ distribution are presented in Figure 5, and we can see that our method maintains strong performance, suggesting that the constraints of Assumption 1 can be substantially relaxed. Additional simulation results can be found in the Appendix A.

**Sensitivity Analysis with respect to smoothness parameter $\gamma$.** We analyze the impact of smoothness parameter $\gamma$ on our proposed algorithm. Specifically, the detailed experimental setup follows the default configuration, and $\gamma$ is set within the range $[2.5 \times 10^{-5}, 1 \times 10^{-3}]$. The simulations are replicated 30 times, and the numerical results of estimation error and four estimated signal strengths are presented in Figure 6. From Figure 6, it's evident that the choice of $\gamma$ is relatively insensitive in the sense that the estimation errors and the estimated strengths of the signals under different $\gamma$s are very close. Furthermore, as $\gamma$ increases, the estimation accuracy experiences a minor decline, but it remains within an acceptable range. See Appendix A for simulation results of Signal 3 and Signal 4.

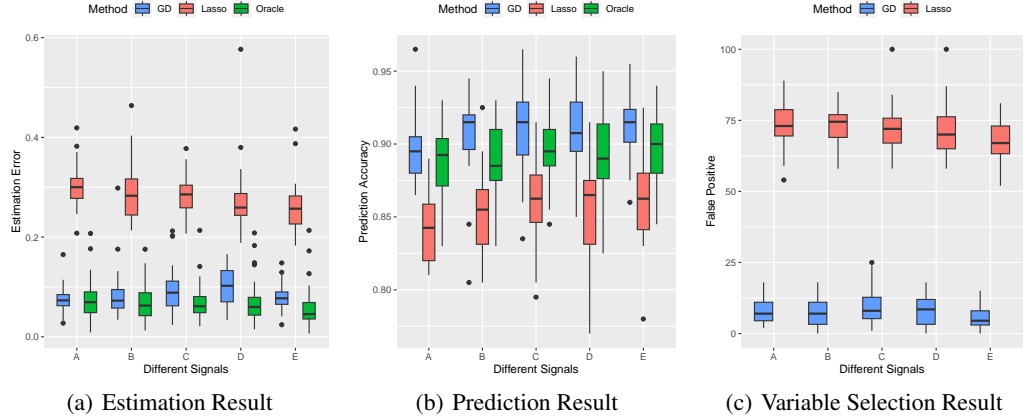

(a) Estimation Result      (b) Prediction Result      (c) Variable Selection Result

Figure 5: Performance on complex signal structure under $t(3)$ distribution. The boxplots are depicted based on 30 runs.

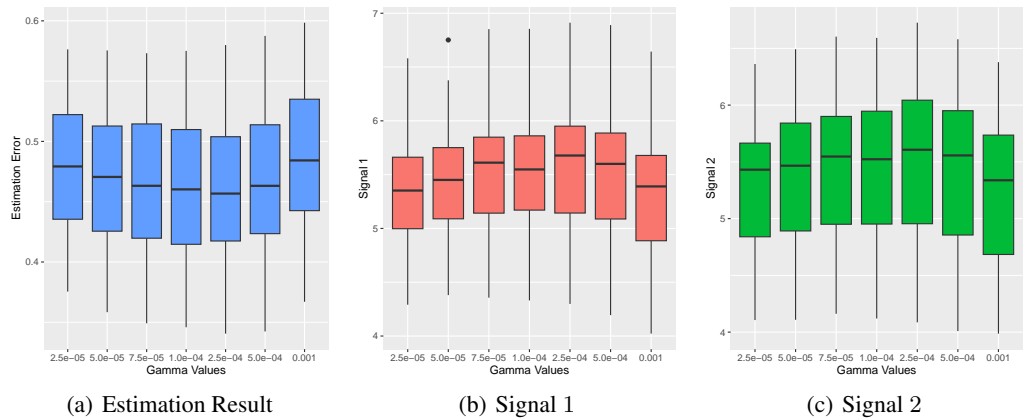

(a) Estimation Result      (b) Signal 1      (c) Signal 2

Figure 6: Sensitivity analysis of smoothness parameter $\gamma$. The boxplots are depicted based on 30 runs. The estimation error is calculated by $\|\boldsymbol{\beta}_t - \boldsymbol{\beta}^*\|/\|\boldsymbol{\beta}^*\|$.

## 5   Conclusion

In this paper, we leverage over-parameterization to design an unregularized gradient-based algorithm for SVM and provide theoretical guarantees for implicit regularization. We employ Nesterov's method to smooth the re-parameterized hinge loss function, which solves the difficulty of non-differentiability and improves computational efficiency. Note that our theory relies on the incoherence of the design matrix. It would be interesting to explore to what extent these assumptions can be relaxed, which is a topic of future work mentioned in other studies on implicit regularization. It is also promising to consider extending the current study to nonlinear SVMs, potentially incorporating the kernel technique to delve into the realm of implicit regularization in nonlinear classification. In summary, this paper not only provides novel theoretical results for over-parameterized SVMs but also enriches the literature on high-dimensional classification with implicit regularization.

## 6   Acknowledgements

The authors sincerely thank the anonymous reviewers, AC, and PCs for their valuable suggestions that have greatly improved the quality of our work. The authors also thank Professor Shaogao Lv for the fruitful and valuable discussions at the very beginning of this work. Dr. Xin He's and Dr. Yang Bai's work is supported by the Program for Innovative Research Team of Shanghai University of Finance and Economics and the Shanghai Research Center for Data Science and Decision Technology.

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

# Appendix

The appendix is organized as follows.

In Appendix A, we provide supplementary explanations and additional experimental results for the numerical study.

In Appendix B, we discuss the performance of our method using other data generating schemes.

In Appendix C, we characterize the dynamics of the gradient descent algorithm for minimizing the Hadamard product re-parametrized smoothed hinge loss $\mathcal{E}^*_{\mathcal{Z}^n,\gamma}$.

In Appendix D, we prove the main results stated in the paper.

In Appendix E, we provide the proof of propositions and technical lemmas in Section 3.

## A  Additional Experimental Results

### A.1  Effects of Small Initialization on Error Components

We present a comprehensive analysis of the impact of initialization on the error components, as depicted in Figure 7. Our results demonstrate that, while the initialization size $\alpha$ does not alter the error trajectory, it significantly affects the error bounds. Numerically, when the initialization $\alpha$ is $10^{-4}$, the maximum absolute value of the error components is around $3 \times 10^{-5}$, which is bounded by the initialization size $10^{-4}$. Similarly, when the initialization is $10^{-10}$, the maximum absolute value of the error components is around $7.5 \times 10^{-14}$, which is bounded by the initialization size $10^{-10}$. The same result is obtained for the other two initialization sizes. The specific numerical results we obtained validate the conclusions drawn in Proposition 1, as the error term satisfies

$$\|\boldsymbol{\beta}_t \odot \mathbf{1}_{S_1^c}\|_\infty = \|\mathbf{w}_t \odot \mathbf{w}_t \odot \mathbf{1}_{S_1^c} - \mathbf{v}_t \odot \mathbf{v}_t \odot \mathbf{1}_{S_1^c}\|_\infty \lesssim (\sqrt{\alpha})^2 = \alpha.$$

### A.2  True Negative Results

In the main text, we could not include the True Negative error figures in 2, 3, 4, and 5 due to space constraints. However, these figures are provided in Figure 8. We observe that both the gradient descent estimator and the Lasso estimator effectively detect the true signals across different settings.

### A.3  Additional Results under Uniform Distribution

As previously described, we sample $\mathbf{X}$ from a $[-1, 1]$ uniform distribution and set other parameters as follows: true signal structures are $\mathbf{A} = (5, 6, 7, 8)$, $\mathbf{B} = (4, 6, 8, 9)$, $\mathbf{C} = (3, 6, 9, 10)$, $\mathbf{D} = (2, 6, 10, 11)$, and $\mathbf{E} = (1, 6, 11, 12)$. The number of signals is $s = 4$, the sample size is $n = 200$, dimension is $p = 400$, step size is $\eta = 0.5$, smoothness parameter is $\gamma = 10^{-4}$, and the initialization size is $\alpha = 10^{-8}$. The experimental results are summarized in Figure 9. As depicted in Figure 9, the gradient descent (GD) estimator consistently outperforms the Lasso estimator in terms of estimation accuracy and variable selection error. Both methods slightly surpass the oracle estimator in terms of prediction. However, this advantage mainly stems from the inevitable overestimation of the number of signals in the estimates by both the GD and Lasso methods.

### A.4  Additional Results of Sensitivity Analysis

Within our sensitivity analysis, the simulation results for Signal 3 and Signal 4 are presented in Figure 10. We observe that the estimated strengths of the two signals exhibit similar distributions across different values of the smoothness parameter $\gamma$.

## B  Other Data Generating Schemes

In section 4, the scheme for generating $y$ in our original submission is based on a treatment similar to those in [41, 42], where the task of variable selection for the support vector machine is considered.

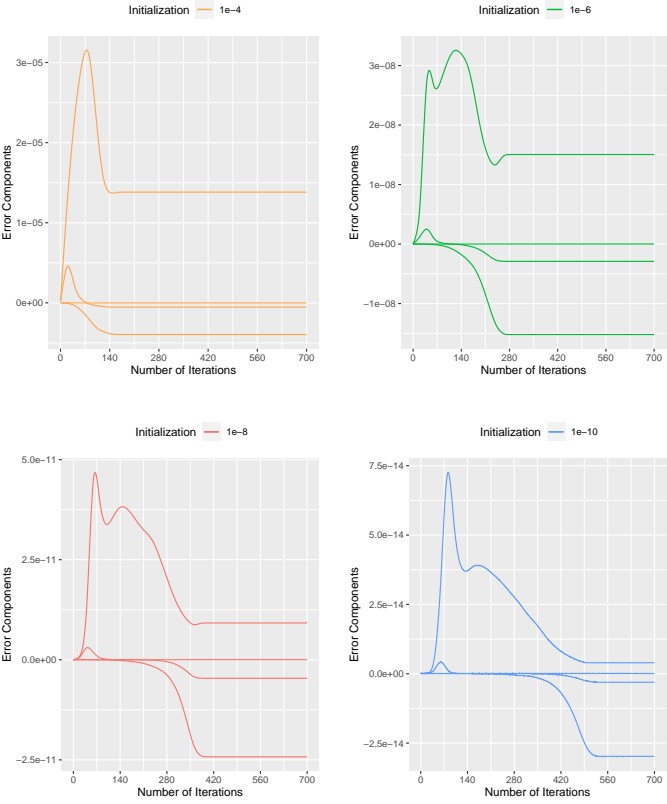

Figure 7: Effects of small initialization $\alpha$ on error components.

In this section, we also introduce other generating schemes as adopted in [32, 15]. Specifically, we generate random data based on the following two models.

- Model 1: $\mathbf{x} \sim MN(\mathbf{0}_p, \mathbf{\Sigma})$, $\mathbf{\Sigma} = (\sigma_{ij})$ with nonzero elements $\sigma_{ij} = 0.4^{|i-j|}$ for $1 \leq i, j \leq p$, $P(y = 1|\mathbf{x}) = \Phi(\mathbf{x}^T \boldsymbol{\beta}^*)$, where $\Phi(\cdot)$ is the cumulative density function of the standard normal distribution, $\boldsymbol{\beta}^* = (1.1, 1.1, 1.1, 1.1, 0, \dots, 0)^T$ and $s = 4$.

- Model 2: $P(y = 1) = P(y = -1) = 0.5$, $\mathbf{x}|(y = 1) \sim MN(\boldsymbol{\mu}, \mathbf{\Sigma})$, $\mathbf{x}|(y = -1) \sim MN(-\boldsymbol{\mu}, \mathbf{\Sigma})$, $s = 5$, $\boldsymbol{\mu} = (0.1, 0.2, 0.3, 0.4, 0.5, 0, \dots, 0)^T$, $\mathbf{\Sigma} = (\sigma_{ij})$ with diagonal entries equal to 1, nonzero entries $\sigma_{ij} = -0.2$ for $1 \leq i \neq j \leq s$ and other entries equal to 0. The bayes rule is $\text{sign}(1.39\mathbf{x}_1 + 1.47\mathbf{x}_2 + 1.56\mathbf{x}_3 + 1.65\mathbf{x}_4 + 1.74\mathbf{x}_5)$ with bayes error 6.3%.

We follow the default parameter setup from Section 4, and the experiments are repeated 30 times. The averaged estimation and prediction results are presented in Figure 11. From Figure 11, it's evident that the GD estimator closely approximates the oracle estimator in terms of estimation error and prediction accuracy.

## C    Gradient Descent Dynamics

First, let's recall the gradient descent updates. We over-parameterize $\boldsymbol{\beta}$ by writing it as $\mathbf{w} \odot \mathbf{w} - \mathbf{v} \odot \mathbf{v}$, where $\mathbf{w}$ and $\mathbf{v}$ are $p \times 1$ vectors. We then apply gradient descent to the following optimization problem:

$$\min_{\mathbf{w}, \mathbf{v}} \max_{\boldsymbol{\mu} \in \mathcal{P}_1} \left\{ \frac{1}{n} \sum_{i=1}^n \left(1 - y_i \mathbf{x}_i^T (\mathbf{w} \odot \mathbf{w} - \mathbf{v} \odot \mathbf{v}) \right) \mu_i - d_\gamma(\boldsymbol{\mu}) \right\},$$

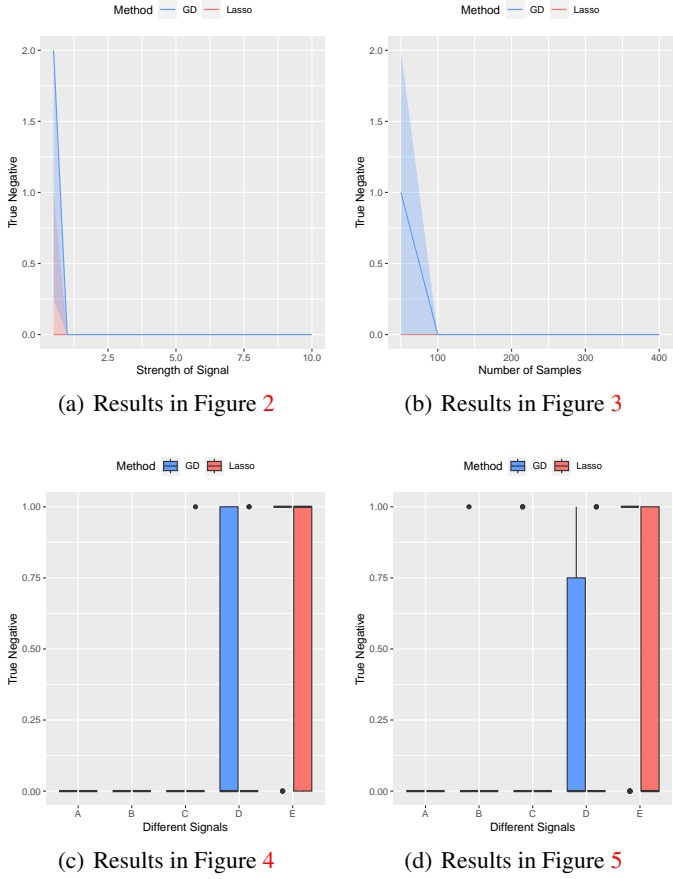

(a) Results in Figure 2

(b) Results in Figure 3

(c) Results in Figure 4

(d) Results in Figure 5

Figure 8: The True Negative results correspond to the settings illustrated in Figures 2, 3, 4 and 5, respectively.

where $d_\gamma(\boldsymbol{\mu}) = \frac{\gamma}{2}\|\boldsymbol{\mu}\|^2$. The gradient descent updates with respect to $\mathbf{w}$ and $\mathbf{v}$ are given by

$$\mathbf{w}_{t+1} = \mathbf{w}_t + 2\eta\frac{1}{n}\sum_{i=1}^{n} y_i\mu_{t,i}\mathbf{x}_i \odot \mathbf{w}_t \quad and \quad \mathbf{v}_{t+1} = \mathbf{v}_t - 2\eta\frac{1}{n}\sum_{i=1}^{n} y_i\mu_{t,i}\mathbf{x}_i \odot \mathbf{v}_t.$$

For the sake of convenience, let $\mathcal{G}_t \in \mathbb{R}^p$ represent the gradients in the form of $\mathcal{G}_t = n^{-1}\mathbf{X}^T\mathbf{Y}\boldsymbol{\mu}_t$, where $\mathbf{Y}$ is a diagonal matrix composed of the elements of $y$. Consequently, the $i$-th element of $\mathcal{G}_t$ can be expressed as $G_{t,i} = n^{-1}\sum_{k=1}^{n} \mu_{t,k}y_k x_{ki}$, where $\mu_{t,k} = \mathrm{median}(0, (1 - y_k\mathbf{x}_k^T\boldsymbol{\beta}_t)/\gamma n, 1)$. Subsequently, the updating rule can be rephrased as follows:

$$\mathbf{w}_{t+1} = \mathbf{w}_t + 2\eta\mathbf{w}_t \odot \mathcal{G}_t \quad and \quad \mathbf{v}_{t+1} = \mathbf{v}_t - 2\eta\mathbf{v}_t \odot \mathcal{G}_t. \tag{7}$$

Furthermore, the error parts of $\mathbf{w}_t$ and $\mathbf{v}_t$ are denoted by $\mathbf{w}_t \odot \mathbf{1}_{S_1^c}$ and $\mathbf{v}_t \odot \mathbf{1}_{S_1^c}$, which include both weak signal parts and pure error parts. In addition, strong signal parts of $\mathbf{w}_t$ and $\mathbf{v}_t$ are denoted by $\mathbf{w}_t \odot \mathbf{1}_{S_1}$ and $\mathbf{v}_t \odot \mathbf{1}_{S_1}$, respectively.

We examine the dynamic changes of error components and strong signal components in two stages. Without loss of generality, we focus on analyzing entries $i \in S_1$ where $\beta_i^* > 0$. The analysis for the case where $\beta_i^* < 0$ and $i \in S_1$ is similar and therefore not presented here. Specifically, within **Stage One**, we can ensure that with a high probability, for $t \geq T_0 = 2\log(m/\alpha^2)/(\eta m)$, the following

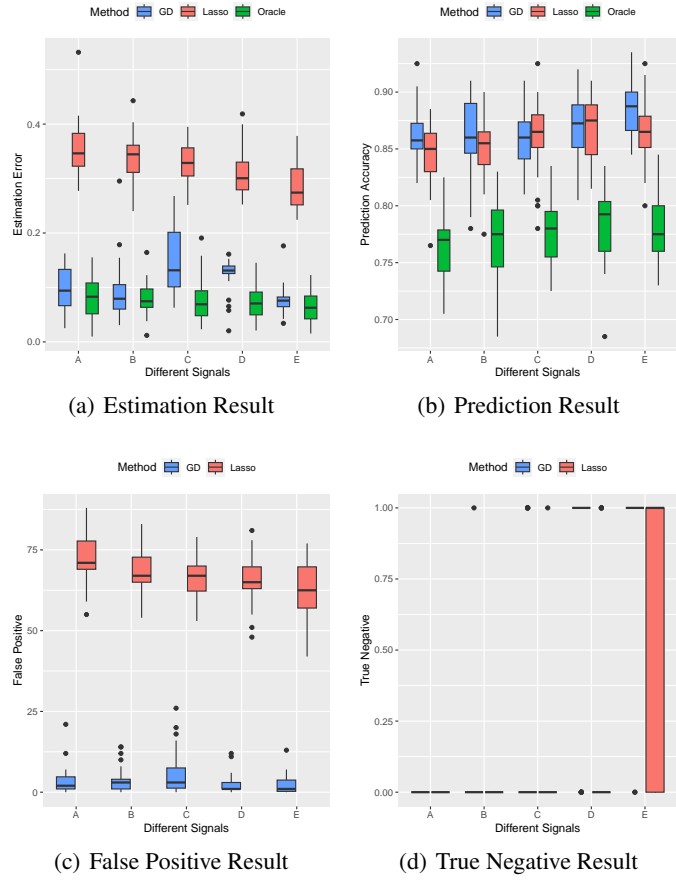

(a) Estimation Result

(b) Prediction Result

(c) False Positive Result

(d) True Negative Result

Figure 9: Performance on complex signal structure under $[-1, 1]$ uniform distribution. The boxplots are depicted based on 30 runs.

results hold under Assumptions 1 and 2:

$$
\textit{strong signal dynamics:} \quad \beta_{t,i} \geq \min\left\{\frac{\beta_i^*}{2}, \left(1 + \frac{\eta\beta_i^*}{\gamma n}\right)^t \alpha^2 - \alpha^2\right\} \quad \textit{for } i \in S_1, \beta_i^* > 0,
$$

$$
\max\left\{\|\mathbf{w}_t \odot \mathbf{1}_{S_1}\|_\infty, \|\mathbf{v}_t \odot \mathbf{1}_{S_1}\|_\infty\right\} \leq \alpha^2 \quad \textit{for } i \in S_1,
$$

$$
\textit{error component dynamics:} \quad \max\left\{\|\mathbf{w}_t \odot \mathbf{1}_{S_1^c}\|_\infty, \|\mathbf{v}_t \odot \mathbf{1}_{S_1^c}\|_\infty\right\} \leq \sqrt{\alpha} \quad \textit{for } i \in S_1^c.
$$

(8)

From (8), we can observe that for $t \geq T_0$, the iterate $(\mathbf{w}_t, \mathbf{v}_t)$ reaches the desired performance level. Specifically, each component $\beta_{t,i}$ of the positive strong signal part $\boldsymbol{\beta}_t \odot \mathbf{1}_{S_1}$ increases exponentially in $t$ until it reaches $\beta_i^*/2$. Meanwhile, the weak signal and error part $\boldsymbol{\beta}_t \odot \mathbf{1}_{S_1^c}$ remains bounded by $\mathcal{O}(\alpha)$. This observation highlights the significance of small initialization size $\alpha$ for the gradient descent algorithm, as it restricts the error term. A smaller initialization leads to better estimation but with the trade-off of a slower convergence rate.

After each component $\beta_{t,i}$ of the strong signal reaches $\beta_i^*/2$, ***Stage Two*** starts. In this stage, $\beta_{t,i}$ continues to grow at a slower rate and converges to the true parameter $\beta_i^*$. After this time, after $3\log(m/\alpha^2)/(\eta m)$ iterations, $\beta_{t,i}$ of the strong signal enters a desired interval, which is within a distance of $C_\epsilon\sqrt{\log p/n}$ from the true parameter $\beta_i^*$. Simultaneously, the error term $\boldsymbol{\beta}_t \odot \mathbf{1}_{S_1^c}$ remains bounded by $\mathcal{O}(\alpha)$ until $\mathcal{O}(\log(1/\alpha)/(\eta\log n))$ iterations.

We summarize the dynamic analysis results described above in Proposition 1 and Proposition 2. By combining the results from these two propositions, we can obtain the proof of Theorem 2 in the subsequent subsection.

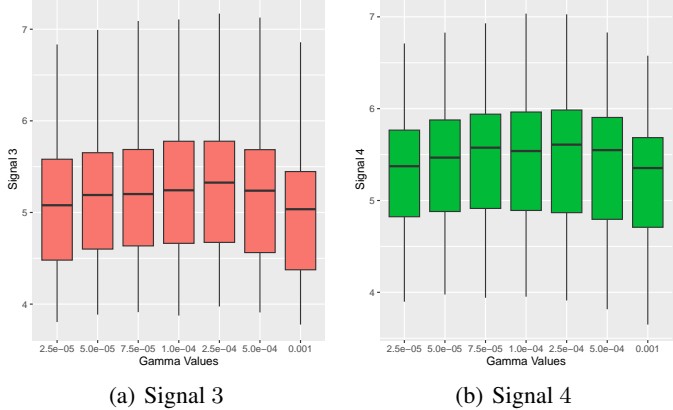

(a) Signal 3                    (b) Signal 4

Figure 10: Sensitivity analysis of smoothness parameter $\gamma$. The boxplots are depicted based on 30 runs.

## D  Proof of Theorem 2

*Proof.* We first utilize Proposition 1 to upper bound the error components $\boldsymbol{\beta}_t \odot \mathbf{1}_{S_1^c}$. By Proposition 1, we are able to control the error parts of the same order as the initialization size $\alpha$ within the time interval $0 \le t \le T^* = \log(1/\alpha)/(4\sigma\eta \log n)$. Thus, by direct computation, we have

$$
\begin{aligned}
\|\boldsymbol{\beta}_t \odot \mathbf{1}_{S_1^c} - \boldsymbol{\beta}^* \odot \mathbf{1}_{S_1^c}\|^2 &= \|\boldsymbol{\beta}_t \odot \mathbf{1}_{S_1^c} + (\boldsymbol{\beta}_t \odot \mathbf{1}_{S_2} - \boldsymbol{\beta}^* \odot \mathbf{1}_{S_2})\|^2 \\
&\overset{(i)}{\le} \|\boldsymbol{\beta}^* \odot \mathbf{1}_{S_2}\|^2 + p\|\mathbf{w}_t \odot \mathbf{w}_t \odot \mathbf{1}_{S_1^c} - \mathbf{v}_t \odot \mathbf{v}_t \odot \mathbf{1}_{S_1^c}\|_\infty^2 \qquad (9) \\
&\overset{(ii)}{\le} C_w^2 \cdot \frac{s_2 \log p}{n} + 2C_\alpha^4 \cdot \frac{1}{p},
\end{aligned}
$$

where $(i)$ is based on the relationship between $\ell_2$ norm and $\ell_\infty$ norm and $(ii)$ follows form the results in Proposition 1. As for the strong signal parts, by Proposition 2, that with probability at least $1 - c_1 n^{-1} - c_2 p^{-1}$, we obtain

$$
\|\boldsymbol{\beta}_t \odot \mathbf{1}_{S_1} - \boldsymbol{\beta}^* \odot \mathbf{1}_{S_1}\|^2 \le s_1 \|\boldsymbol{\beta}_t \odot \mathbf{1}_{S_1} - \boldsymbol{\beta}^* \odot \mathbf{1}_{S_1}\|_\infty^2 \le C_\epsilon^2 \cdot \frac{s_1 \log p}{n}. \qquad (10)
$$

Finally, combining (9) and (10), with probability at least $1 - c_1 n^{-1} - c_2 p^{-1}$, for any $t$ that belongs to the interval

$$
[5\log(m/\alpha^2)/(\eta m), \log(1/\alpha)/(4\sigma\eta \log n)],
$$

it holds that

$$
\begin{aligned}
\|\boldsymbol{\beta}_{T_1} - \boldsymbol{\beta}^*\|^2 &= \|\boldsymbol{\beta}_t \odot \mathbf{1}_{S_1^c} - \boldsymbol{\beta}^* \odot \mathbf{1}_{S_1^c}\|^2 + \|\boldsymbol{\beta}_t \odot \mathbf{1}_{S_1} - \boldsymbol{\beta}^* \odot \mathbf{1}_{S_1}\|^2 \\
&\le C_\epsilon^2 \cdot \frac{s_1 \log p}{n} + C_w^2 \cdot \frac{s_2 \log p}{n} + 2C_\alpha^4 \cdot \frac{1}{p}.
\end{aligned}
$$

Since $p$ is much larger than $n$, the last term, $2C_\alpha^4/p$, is negligible. Considering the constants $C_w$, $C_\alpha$ and $C_\epsilon$, we finally obtain the error bound of gradient descent estimator in terms of $\ell_2$ norm. Therefore, we conclude the proof of Theorem 2.  $\square$

## E  Proof of Propositions 1 and 2

In this section, we provide the proof for the two propositions mentioned in Section 3. First, we introduce some useful lemmas, which are about the coherence of the design matrix $\mathbf{X}$ and the upper bound of sub-Gaussian random variables.

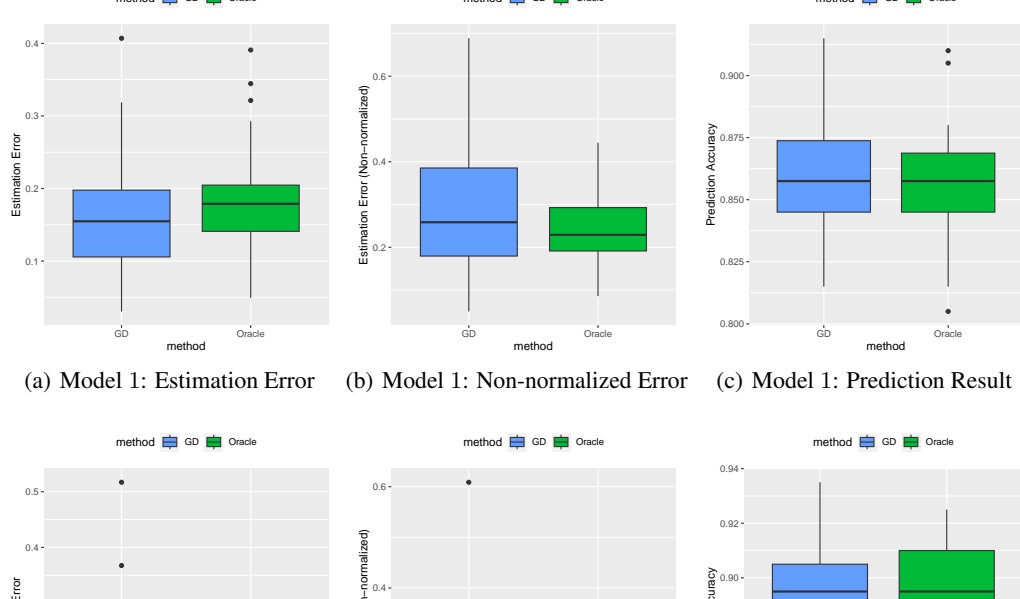

(a) Model 1: Estimation Error    (b) Model 1: Non-normalized Error    (c) Model 1: Prediction Result

(d) Model 2: Estimation Error    (e) Model 2: Non-normalized Error    (f) Model 2: Prediction Result

Figure 11: Estimation error and prediction performance on Model 1 and Model 2. The boxplots are depicted based on 30 runs. The non-normalized error is calculated by $\|\boldsymbol{\beta}_t - \boldsymbol{\beta}^*\|/\|\boldsymbol{\beta}^*\|$.

### E.1 Technical Lemmas

**Lemma 1.** *(General Hoeffding's inequality) Suppose that $\theta_1, \cdots, \theta_m$ are independent mean-zero sub-Gaussian random variables and $\boldsymbol{\mu} = (\mu_1, \cdots, \mu_m) \in \mathbb{R}^m$. Then, for every $t > 0$, we have*

$$\mathbb{P}\left[\left|\sum_{i=1}^{m} \mu_i \theta_i\right| \geq t\right] \leq 2 \exp\left(-\frac{ct^2}{d^2\|\boldsymbol{\mu}\|^2}\right),$$

*where $d = \max_i \|\theta_i\|_{\psi_2}$ and $c$ is an absolute constant.*

*Proof.* General Hoeffding's inequality can be proofed directly by the independence of $\theta_1, \cdots, \theta_m$ and the sub-Gaussian property. The detailed proof is omitted here. $\square$

**Lemma 2.** *Suppose that $\mathbf{X}/\sqrt{n}$ is a $n \times p$ $\ell_2$-normalized matrix satisfying $\delta$-incoherence; that is $1/n|\langle \mathbf{x}_i \odot \mathbf{1}_K, \mathbf{x}_i \odot \mathbf{1}_K\rangle| \leq \delta, i \neq j$ for any $K \subseteq [n]$. Then, for $s$-sparse vector $\mathbf{z} \in \mathbb{R}^p$, we have*

$$\left\|\left(\frac{1}{n}\mathbf{X}_K^T\mathbf{X}_K - \mathbf{I}\right)\mathbf{z}\right\|_{\infty} \leq \delta s\|\mathbf{z}\|_{\infty},$$

*where $\mathbf{X}_K$ denotes the $n \times p$ matrix whose $i$-th column is $\mathbf{x}_i$ if $i \in K$ and $\mathbf{0}_{p\times 1}$ otherwise.*

*Proof.* The proof of Lemma 2 is similar to Lemma 2 in [23]. According to the condition

$$\frac{1}{n}|\langle \mathbf{x}_i \odot \mathbf{1}_K, \mathbf{x}_i \odot \mathbf{1}_K\rangle| \leq \delta, i \neq j, K \subseteq [n],$$

we can verify that for any $i \in [p]$,

$$\left| \left( \frac{1}{n} \mathbf{X}_K^T \mathbf{X}_K \mathbf{z} \right)_i - z_i \right| \le \delta s \|\mathbf{z}\|_\infty.$$

Therefore,

$$\left\| \left( \frac{1}{n} \mathbf{X}_K^T \mathbf{X}_K - \mathbf{I} \right) \mathbf{z} \right\|_\infty \le \delta s \|\mathbf{z}\|_\infty.$$

$\square$

### E.2   Proof of Proposition 1

*Proof.* Here we prove Proposition 1 by induction hypothesis. It holds that our initializations $\mathbf{w}_0 = \alpha \mathbf{1}_{p \times 1}$ and $\mathbf{v}_0 = \alpha \mathbf{1}_{p \times 1}$ satisfy our conclusion given in Proposition 1. As we initialize $w_{0,i}$ and $v_{0,i}$ for any fixed $i \in S_1^c$ with

$$|w_{0,i}| = \alpha \le \sqrt{\alpha} \quad and \quad |v_{0,i}| = \alpha \le \sqrt{\alpha},$$

Proposition 1 holds when $t = 0$. In the following, we show that for any $t^*$ with $0 \le t^* \le T^* = \log(1/\alpha)/(4\sigma\eta \log n)$, if the conclusion of Proposition 1 stands for all $t$ with $0 \le t < t^*$, then it also stands at the $(t^* + 1)$-th step. From the gradient descent updating rule given in (7), the updates of the weak signals and errors $w_{t,i}, v_{t,i}$ for any fixed $i \in S_1^c$ are obtained as follows,

$$w_{t+1,i} = w_{t,i}(1 + 2\eta G_{t,i}) \quad and \quad v_{t+1,i} = v_{t,i}(1 - 2\eta G_{t,i}).$$

Recall that $|\mu_{t,k}| \le 1$ for $k = 1, \dots, n$, for any fixed $i \in S_1^c$, then we have

$$|G_{t,i}| = \left| \frac{1}{n} \sum_{k=1}^n \mu_{t,k} y_k x_{ki} \right| \le \frac{1}{n} \sum_{k=1}^n |x_{ki}|.$$

For ease of notation, we denote $M_i = n^{-1} \sum_{k=1}^n |x_{ki}|$ and $M = \max_{i \in S_1^c} M_i$. Then we can easily bound the weak signals and errors as follows,

$$\|\mathbf{w}_{t+1} \odot \mathbf{1}_{S_1^c}\|_\infty \le (1 + 2\eta M)\|\mathbf{w}_t \odot \mathbf{1}_{S_1^c}\|_\infty \quad and \quad \|\mathbf{v}_{t+1} \odot \mathbf{1}_{S_1^c}\|_\infty \le (1 + 2\eta M)\|\mathbf{v}_t \odot \mathbf{1}_{S_1^c}\|_\infty. \tag{11}$$

By General Hoeffding's inequality of sub-Gaussian random variables, with probability at least $1 - cn^{-n}$, we obtain $M_i \le \sigma\sqrt{\log n}, i \in S_1^c$, where $c$ is a constant. Since $p$ is much larger than $n$, the inequality $(1 - cn^{-n})^p > 1 - cn^{-1}$ holds. Then, with probability at least $1 - cn^{-1}$, where $c$ is a constant, we have $M \le \sigma\sqrt{\log n}$.

Combined (11) and the bound of $M$, we then have

$$
\begin{aligned}
\|\mathbf{w}_{t^*+1} \odot \mathbf{1}_{S_1^c}\|_\infty &\le (1 + 2\eta M)^{t^*+1} \|\mathbf{w}_0 \odot \mathbf{1}_{S_1^c}\|_\infty \\
&= \exp((t^* + 1)\log(1 + 2\eta M))\alpha \\
&\overset{(i)}{\le} \exp(T^* \log(1 + 2\eta M))\alpha \\
&\overset{(ii)}{\le} \exp(T^* \cdot 2\eta M)\alpha \\
&\overset{(iii)}{\le} \exp((1/2)\log(1/\alpha))\alpha = \sqrt{\alpha},
\end{aligned}
$$

where $(i)$ and $(ii)$ follow from $t^* + 1 \le T^*$ and $\log(1 + x) < x$ when $x > 0$, respectively. Moreover, $(iii)$ is based on the definition of $T^*$. Similarly, we can prove that $\|v_{t^*} \odot \mathbf{1}_{S_1^c}\|_\infty \le \sqrt{\alpha}$. Thus, the induction hypothesis also holds for $t^* + 1$. Since $t^* < T^*$ is arbitrarily chosen, we complete the proof of Proposition 1. $\square$

### E.3   Proof of Proposition 2

*Proof.* In order to analyze strong signals: $\beta_{t,i} = w_{t,i}^2 - v_{t,i}^2$ for any fixed $i \in S_1$, we focus on the dynamics $w_{t,i}^2$ and $v_{t,i}^2$. Following the updating rule (7), we have

$$
\begin{aligned}
w_{t+1,i}^2 &= w_{t,i}^2 + 4\eta w_{t,i}^2 G_{t,i} + 4\eta^2 w_{t,i}^2 G_{t,i}^2, \\
v_{t+1,i}^2 &= v_{t,i}^2 - 4\eta v_{t,i}^2 G_{t,i} + 4\eta^2 v_{t,i}^2 G_{t,i}^2.
\end{aligned}
\tag{12}
$$

Without loss of generality, here we just analyze entries $i \in S_1$ with $\beta_i^* > 0$. Analysis for the negative case with $\beta_i^* < 0, i \in S_1$ is almost the same and is thus omitted. We divide our analysis into two stages, as discussed in Appendix C.

Specifically, in **Stage One**, since we choose the initialization as $\mathbf{w}_0 = \mathbf{v}_0 = \alpha \mathbf{1}_{p \times 1}$, we obtain $\beta_{0,i} = 0$ for any fixed $i \in S_1$, we show after $2 \log(\beta_i^*/\alpha^2)/\eta \beta_i^*$ iterations, the gradient descent coordinate $\beta_{t,i}$ will exceed $\beta_i^*/2$. Therefore, all components of strong signal part will exceed half of the true parameters after $2 \log(m/\alpha^2)/(\eta m)$ iterations. Furthermore, we calculate the number of iterations required to achieve $\beta_{t,i} \geq \beta_i^* - \epsilon$, where $\epsilon = C_\epsilon \sqrt{\log p / n}$, in **Stage Two**. Thus, we conclude the proof of Proposition 2.

**Stage One.** First, we introduce some new notations. In $t$-th iteration, according to the values of $\mu_{t,i}$, we divide the set $[n]$ into three parts, $K_{t,1} = \{i \in [n] : \mu_{t,i} = 1\}$, $K_{t,2} = \{i \in [n] : 0 < \mu_{t,i} < 1\}$ and $K_{t,3} = \{i \in [n] : \mu_{t,i} = 0\}$, with cardinalities of $k_{t,1}$, $k_{t,2}$ and $k_{t,3}$, respectively. From [28, 45], most elements of $\boldsymbol{\mu}_t$ are 0 or 1 and the proportion of these elements will rapidly increase with the decreasing of $\gamma$, which means that $k_{t,2} \ll (k_{t,1} + k_{t,3})$ controlling $\gamma$ in a small level.

According to the updating formula of $w_{t+1,i}^2$ in (12), we define the following form of element-wise bound $\xi_{t,i}$ for any fixed $i \in S_1$ as

$$
\begin{aligned}
\xi_{t,i} :&= 1 - \frac{w_{t,i}^2}{w_{t+1,i}^2} \left( 1 - \frac{4\eta}{\gamma n}(\beta_{t,i} - \beta_i^*) \right) \\
&= \frac{1}{w_{t+1,i}^2} \left( w_{t+1,i}^2 - w_{t,i}^2 + \frac{4\eta}{\gamma n} w_{t,i}^2 (\beta_{t,i} - \beta_i^*) \right) \\
&= \frac{w_{t,i}^2}{w_{t+1,i}^2} \left( 4\eta \left( G_{t,i} + \frac{1}{\gamma n}(\beta_{t,i} - \beta_i^*) \right) + 4\eta^2 G_{t,i}^2 \right).
\end{aligned}
\tag{13}
$$

Rewriting (13), we can easily get that

$$
w_{t+1,i}^2 = w_{t,i}^2 \left( 1 - \frac{4\eta}{\gamma n}(\beta_{t,i} - \beta_i^*) \right) + \xi_{t,i} w_{t+1,i}^2.
\tag{14}
$$

From (14), it is obvious that if the magnitude of $\xi_{t,i}$ is sufficiently small, we can easily conclude that $w_{t+1,i}^2 \geq w_{t,i}^2$. Therefore, our goal is to evaluate the magnitude of $\xi_{t,i}$ in (13) for any fixed $i \in S_1$. First, we focus on the term $G_{t,i} + (\beta_{t,i} - \beta_i^*)/(\gamma n)$ in (13). In particular, recall the definition of $G_{t,i}$ and we expand $G_{t,i} + (\beta_{t,i} - \beta_i^*)/(\gamma n)$ in the following form

$$
\begin{aligned}
G_{t,i} + \frac{1}{\gamma n}(\beta_{t,i} - \beta_i^*) &= \frac{1}{n} \left( \sum_{k \in K_{t,1}} y_k x_{ki} + \sum_{k \in K_{t,2}} y_k x_{ki} \frac{1 - y_k \mathbf{x}_k^T \boldsymbol{\beta}_t}{n\gamma} \right) + \frac{1}{\gamma n}(\beta_{t,i} - \beta_i^*) \\
&= \frac{1}{n} \left( \sum_{k \in K_{t,1}} y_k x_{ki} + \sum_{k \in K_{t,2}} \frac{y_k x_{ki}}{n\gamma} \right) - \frac{1}{\gamma n^2} \left( \sum_{k \in K_{t,2}} x_{k,i} \mathbf{x}_k^T \boldsymbol{\beta}_t - n(\beta_{t,i} - \beta_i^*) \right) \\
&= \frac{1}{n} \left( \sum_{k \in K_{t,1}} y_k x_{ki} + \sum_{k \in K_{t,2}} \frac{(y_k - \mathbf{x}_k^T \boldsymbol{\beta}^*) x_{ki}}{n\gamma} \right) \\
&\quad - \frac{1}{\gamma n} \left( \frac{1}{n} \sum_{k \in K_{t,2}} x_{k,i} \mathbf{x}_k^T (\boldsymbol{\beta}_t - \boldsymbol{\beta}^*) - (\beta_{t,i} - \beta_i^*) \right) \\
&\widehat{=} (I_1) + (I_2).
\end{aligned}
$$

For the term $(I_1)$, by the condition $k_{t,2} \ll k_{t,1}$ and General Hoeffding's inequality, we have

$$
\frac{1}{n} \left| \sum_{k \in K_{t,1}} y_k x_{ki} + \sum_{k \in K_{t,2}} \frac{(y_k - \mathbf{x}_k^T \boldsymbol{\beta}^*) x_{ki}}{n\gamma} \right| \leq \frac{1}{n} \sigma \sqrt{n \log p} = \sigma \sqrt{\frac{\log p}{n}},
\tag{15}
$$

holds with probability at least $1 - cp^{-1}$, where $c$ is a constant.

For the term $(I_2)$, let $\mathbf{X}_{t,2} \in \mathbb{R}^{n \times p}$ denote the matrix whose $k$-th column is $\mathbf{x}_k$ if $k \in K_{t,2}$. Then, we approximate $(I_2)$ based on the assumption on the design matrix $\mathbf{X}$ via Lemma 2,

$$\left\| \frac{1}{n} \mathbf{X}_{t,2}^T \mathbf{X}_{t,2} (\boldsymbol{\beta}_t - \boldsymbol{\beta}^*) - (\boldsymbol{\beta}_t \odot \mathbf{1}_{S_1} - \boldsymbol{\beta}^* \odot \mathbf{1}_{S_1}) \right\|_\infty \lesssim \delta s \kappa m. \tag{16}$$

By (15), (16), condition of the minimal strength $m \geq \sigma \log p \sqrt{\log p / n}$ and condition $\delta \lesssim 1/(\kappa s \log p)$, we have

$$\left| G_{t,i} + \frac{1}{\gamma n} (\beta_{t,i} - \beta_i^*) \right| \lesssim \frac{m}{\gamma n \log p}. \tag{17}$$

Then, we can bound $G_{t,i}$ through

$$|G_{t,i}| \leq |G_{t,i} + 1/(\gamma n)(\beta_{t,i} - \beta_i^*)| + |1/(\gamma n)(\beta_{t,i} - \beta_i^*)|$$
$$\lesssim \frac{1}{\gamma n} \left( \frac{m}{\log p} + \kappa m \right). \tag{18}$$

Note that for any fixed $i \in S_1$, $G_{t,i} \leq n^{-1} \sum_{k=1}^n |x_{ki}|$. By General Hoeffding's inequality, we have with probability at least $1 - c_1 n^{-1}$, $|G_{t,i}| \lesssim \log n$ and then $|\eta G_{t,i}| < 1$ based on the assumption of $\eta$. Therefore, $w_{t,i}^2 / w_{t+1,i}^2$ is always bounded by some constant greater than 1.

Recalling the definition of element-wise bound $\xi_{t,i}$, and combining (17) and (18), we have that

$$|\xi_{t,i}| \overset{(i)}{\lesssim} \frac{1}{\gamma n} \left( \frac{\eta m}{\log p} + \eta^2 \kappa^2 m^2 \right) \overset{(ii)}{\lesssim} \frac{\eta m}{\gamma n \log p}, \tag{19}$$

where we use the bound of $w_{t,i}^2 / w_{t+1,i}^2$ for any fixed $i \in S_1$ in step$(i)$ and step $(ii)$ is based on $\eta \kappa m \leq m/\log p$ and $\kappa m \lesssim 1$. Thus, we conclude that $\xi_{t,i}$ is sufficiently small for any fixed $i \in S_1$.

Now, for any fixed $i \in S_1$, when $0 \leq \beta_{t,i} \leq \beta_i^*/2$, we can get the increment of $w_{t,i}^2$ for any fixed $i \in S_1$ according to (19),

$$w_{t+1,i}^2 = w_{t,i}^2 \left( 1 - \frac{4\eta}{\gamma n} (\beta_{t,i} - \beta_i^*) \right) \Big/ \left( 1 + c \frac{\eta m}{\gamma n \log p} \right)$$
$$\overset{(i)}{\geq} w_{t,i}^2 \left( 1 + \frac{2\eta \beta_i^*}{\gamma n} \right) \Big/ \left( 1 + c \frac{\eta m}{\gamma n \log p} \right)$$
$$\overset{(ii)}{\geq} w_{t,i}^2 \left( 1 + \frac{\eta \beta_i^*}{\gamma n} \right),$$

where $(i)$ follows from $0 \leq \beta_{t,i} \leq \beta_i^*/2$ and $(ii)$ holds since $m/\log p \lesssim \beta_i^*$. Similarly, we can analyze $v_{t,i}^2$ to get that when $0 \leq \beta_{t,i} \leq \beta_i^*/2$,

$$v_{t+1,i}^2 \leq v_{t,i}^2 \left( 1 - \eta \beta_i^* / \gamma n \right).$$

Therefore, $w_{t,i}^2$ increases at an exponential rate while $v_{t,i}^2$ decreases at an exponential rate. **Stage One** ends when $\beta_{t,i}$ exceeds $\beta_i^*/2$, and our goal is to estimate $t_{i,0}$ that satisfies

$$\beta_{t,i} \geq \left( 1 + \frac{\eta \beta_i^*}{\gamma n} \right)^{t_{i,0}} \alpha^2 - \left( 1 - \frac{\eta \beta_i^*}{\gamma n} \right)^{t_{i,0}} \alpha^2 \geq \beta_i^*/2.$$

Note that $\{v_{t,i}^2\}_{t \geq 0}$ forms a decreasing sequence and thus is bounded by $\alpha^2$. Hence, it suffices to solve the following inequality for $t_{i,0}$,

$$\left( 1 + \frac{\eta \beta_i^*}{\gamma n} \right)^{t_{i,0}} \alpha^2 \geq \beta_i^*/2 + \alpha^2,$$

which is equivalent to obtain $t_{i,0}$ satisfying

$$t_{i,0} \geq T_{i,0} = \log \left( \frac{\beta^*}{2\alpha^2} + 1 \right) \Big/ \log \left( 1 + \frac{\eta \beta_i^*}{\gamma n} \right).$$

For $T_{i,0}$, by direct calculation, we have

$$
T_{i,0} \overset{(i)}{\leq} \log\left(\frac{\beta^*}{2\alpha^2}+1\right)\left(1+\frac{\eta\beta_i^*}{\gamma n}\right)\Big/\left(\frac{\eta\beta_i^*}{\gamma n}\right)
$$

$$
\overset{(ii)}{\leq} 2\log\left(\frac{\beta_i^*}{\alpha^2}\right)\Big/\eta\beta_i^*,
$$

where $(i)$ follows from $x\log x - x + 1 \geq 0$ when $x \geq 0$ and $(ii)$ holds due to $\log(x/2+1) \leq \log x$ when $x \geq 2$ as well as the assumption on $\gamma$ and $\eta\beta_i^* \leq 1$. Thus, we set $t_{i,0} = 2\log(\beta_i^*/\alpha^2)/(\eta\beta_i^*)$ such that for all $t \geq t_{i,0}$, $\beta_{t,i} \geq \beta_i^*/2$ for all $i \in S_1$.

***Stage Two.*** Define $\epsilon = C_\epsilon\sqrt{\log p/n}$ and $a_{i,1} = \lceil\log_2(\beta_i^*/\epsilon)\rceil$. Next, we refine the element-wise bound $\xi_{t,i}$ according to (13). For any fixed $i \in S_1$, if there exists some $a$ such that $1 \leq a \leq a_{i,1}$ and $(1-1/2^a)\beta_i^* \leq \beta_{t,i} \leq (1-1/2^{a+1})\beta_i^*$, then, based on the analytical thinking in ***Stage One***, we can easily deduce that

$$
|\xi_{t,i}| \lesssim \frac{\eta m}{2^a \gamma n \log p}. \tag{20}
$$

Using this element-wise bound (20), we get the increment of $w_{t+1,i}^2$ and decrement of $v_{t+1,i}^2$,

$$
w_{t+1,i}^2 \geq w_{t,i}^2\left(1+\frac{\eta\beta_i^*}{2^a\gamma n}\right) \quad and \quad v_{t+1,i}^2 \leq v_{t,i}^2\left(1-\frac{\eta\beta_i^*}{2^a\gamma n}\right).
$$

We define $t_{i,a}$ as the smallest $t$ such that $\beta_{t+t_{i,a},i} \geq (1-1/2^{a+1})\beta_i^*$. Intuitively, suppose that the current estimate $\beta_{t,i}$ is between $(1-1/2^a)\beta_i^*$ and $(1-1/2^{a+1})\beta_i^*$, we aim to find the number of iterations required for the sequence $\{\beta_{t,i}\}_{t\geq 0}$ to exceed $(1-1/2^{a+1})\beta_i^*$. To obtain a sufficient condition for $t_{i,a}$, we construct the following inequality,

$$
\beta_{t+t_{i,a},i} \geq w_{t,i}^2\left(1+\frac{\eta\beta_i^*}{2^a\gamma n}\right)^{t_{i,a}} - v_{t,i}^2\left(1-\frac{\eta\beta_i^*}{2^a\gamma n}\right)^{t_{i,a}} \geq (1-1/2^{a+1})\beta_i^*.
$$

Similar with ***Stage One***, the sequence $\{v_{t,i}^2\}_{t\geq 0}$ is bounded by $\alpha^2$. Then, it is sufficient to solve the following inequality for $t_{i,a}$,

$$
t_{i,a} :\geq T_{i,a} = \log\left(\frac{(1-1/2^{a+1})\beta_i^* + \alpha^2}{w_{t,i}^2}\right)\Big/\log\left(1+\frac{\eta\beta_i^*}{2^a\gamma n}\right).
$$

To obtain a more precise upper bound of $T_{i,a}$, we have

$$
T_{i,a} = \log\left(\frac{(1-1/2^{a+1})\beta_i^* + \alpha^2}{w_{t,i}^2}\right)\Big/\log\left(1+\frac{\eta\beta_i^*}{2^a\gamma n}\right)
$$

$$
\overset{(i)}{\leq} \log\left(\frac{(1-1/2^{a+1})\beta_i^* + \alpha^2}{(1-1/2^a)\beta_i^*}\right)\Big/\log\left(1+\frac{\eta\beta_i^*}{2^a\gamma n}\right)
$$

$$
\overset{(ii)}{\leq} \log\left(1+\frac{1/2^{a+1}}{1-1/2^a}+\frac{\alpha^2}{(1-1/2^a)\beta_i^*}\right)\left(1+\frac{\eta\beta_i^*}{2^a\gamma n}\right)\Big/\left(\frac{\eta\beta_i^*}{2^a\gamma n}\right),
$$

where $(i)$ is based on the condition $w_{t,i}^2 \geq (1-1/2^a)\beta_i^*$, $(ii)$ stands since $x\log x - x + 1 \geq 0$ when $x \geq 0$. By direct calculation, we obtain

$$
T_{i,a} \overset{(iii)}{\leq} \left(\frac{1/2^{a+1}}{1-1/2^a}+\frac{\alpha^2}{(1-1/2^a)\beta_i^*}\right)\Big/\left(\frac{\eta\beta_i^*}{2^{a+1}\gamma n}\right)
$$

$$
\overset{(iv)}{\leq} \frac{2}{\eta\beta_i^*}+\frac{2^{a+2}\alpha^2}{\eta\beta_i^{*2}}, \tag{21}
$$

where we use $\log(x+1) \leq x$ when $x \geq 0$ in step $(iii)$ and $1-1/2^a \geq 1/2$ in step $(iv)$, respectively. Recall the assumption $a \leq a_{i,1} = \lceil\log_2(\beta_i^*/\epsilon)\rceil$ with $\epsilon = C_\epsilon\sqrt{\log p/n}$, then we get $2^{a+2} \leq$

$4\beta_i^*/\epsilon \leq 4\sqrt{n/\log p}\,\beta_i^*/C_\epsilon$. Moreover, by the assumption on the initialization size $\alpha$, we have $\alpha^2 \leq C_\alpha/p^2$. Thus, we can bound $2^{a+2}\alpha^2/\eta\beta_i^{*2}$ in (21) as

$$\frac{2^{a+2}\alpha^2}{\eta\beta_i^{*2}} \leq \sqrt{\frac{n}{\log p}}\frac{4C_\alpha}{C_\epsilon p^2\eta\beta_i^*} \overset{(i)}{\leq} \frac{1}{\eta\beta_i^*}, \tag{22}$$

where $(i)$ holds when $p^2\log p \geq 4C_\alpha\sqrt{n}/C_\epsilon$. Combined (21) and (22), we calculate the final bound of $T_{i,a}$ as

$$T_{i,a} \leq \frac{3}{\eta\beta_i^*},$$

for any $a \leq a_{i,1}$. If there exists an $1 \leq a \leq a_{i,1}$ such that $\beta_{t,i} \in [(1-1/2^a)\beta_i^*, (1-1/2^{a+1})\beta_i^*]$, after $t_{i,a} \geq 3/\eta\beta_i^*$ iterations, we can guarantee that $\beta_{t,i} \geq (1-1/2^{a+1})\beta_i^*$. Now recalling the definition of $a_{i,1}$, we have $\beta_i^*/2^{a_{i,1}} \leq \epsilon = C_\epsilon\sqrt{\log p/n}$. Therefore, with at most $\sum_{a=0}^{a_{i,1}} T_{i,a}$ iterations, we have $\beta_{t,i} \geq \beta_i^* - \epsilon$. By the assumption of $\alpha$, we obtain

$$\sum_{a=0}^{a_{i,1}} T_{i,a} \leq 2\log(\beta_i^*/\alpha^2)/(\eta\beta_i^*) + 3\left\lceil\log_2\left(\frac{\beta_i^*}{\epsilon}\right)\right\rceil\bigg/(\eta\beta_i^*)$$

$$\leq 5\log\left(\frac{\beta_i^*}{\alpha^2}\right)\bigg/(\eta\beta_i^*).$$

Since the function $\log(x/\alpha^2)/(\eta x)$ is decreasing with respect to $x$, we have after $5\log(m/\alpha^2)/(\eta m)$ iterations that $|\beta_{t,i} - \beta_i^*| \lesssim \sqrt{\log p/n}$ for any fixed $i \in S_1$. Thus, we complete the proof of Proposition 2. $\qquad\square$

