# OpenReview forum: "Implicit Regularization in Over-Parameterized Support Vector Machine"
_NeurIPS.cc/2023/Conference — NeurIPS 2023 poster_

### Official Review · Reviewer_4Xb3 · 2023-07-05

**Soundness:** 2 fair
**Presentation:** 3 good
**Contribution:** 2 fair
**Rating:** 5
**Confidence:** 3

**Summary:**

A regularization-free algorithm for high-dimensional support vector machines (SVMs) is designed by integrating over-parameterization with Nesterov's smoothing method, which induces implicit regularization. An over-parameterized hinge loss function is constructed and true parameters are estimated by leveraging regularization-free gradient descent on it. Nesterov's method enhances the computational efficiency of the algorithm, particularly in terms of determining the stopping criterion and reducing computational complexity. With appropriate choices of initialization, stepsize, and prox-parameter, unregularized gradient descent achieves a near-oracle statistical convergence rate. The theoretical findings are verified through a variety of numerical experiments and the proposed method is compared with explicit  $ l\_1 $ regularization. The advantages of employing implicit regularization via gradient descent subsequent to over-parameterization in sparse SVM are illustrated by the results.

**Strengths:**

Strenth 1: This manuscript presents an interesting finding about the over-parameterized SVM by showing that if we re-parameterize the sparse unknown parameter $\beta$ by $w\odot w-v\odot v$, then Nesterov's smoothing based alternative update of variables leads to the effect of implicit regularization. That is to say, even we did not explicitly impose $l\_1$-norm onto the underlying parameter, the algorithm behaves as if the $l\_1$-norm is implicitly added.

Strength 2: The theoretical analysis is interesting and easy to follow.

Strength 3: The theoretical findings are validated with numerical simulations.

**Weaknesses:**

Weakness 1: Although it is an interesting result for over-parameterized SVM, the proposed algorithm is not directly designed for the original model (2), but its modifed version in Sec. 2.3. It seems that the proposed Algorithm 1 might have some explicit regulrization term in its single iteration. Specifically, the update of $\mu$ is due to Eq. (4) which introduces an extra smoothing term $d\_\gamma(\mu)=\gamma/2\|\mu\|_2^2$ and this $\mu$-update surely affects $w$ and $v$, which may result an regulurization effect like $l\_1$.

Weakness 2: The Assumption 1 requires the design matrix to satisfy the $\delta$-incoherence and the sub-Gaussianality. It seems this assumption is relatively strict for real datasets.

Weakness 3: The empirical effectiveness of the proposed algorithm is insufficiently discussed.

**Questions:**

Question 1: Since the update of $\mu$ in Algorithm 1 is due to Eq. (4) which introduces an extra smoothing term $d\_\gamma(\mu)=\gamma/2\|\mu\|_2^2$ and this $\mu$-update surely affects $w$ and $v$.  Is it possible that this additional smoothing term intruduces some explicit regularization effects?

Question/Suggestion 2: As  Assumption 1 requires the design matrix to satisfy the $\delta$-incoherence and the sub-Gaussianality, it is suggested to discuss whether this assumption holds on real data.

Question 3: Is it possible to discuss the empirical effectiveness of the proposed algorithm?

**Limitations:**

It seems the authors did not sufficiently address the limitation due to the lack of explicit discussion in the main text.

---

> ### Author Rebuttal · Authors · 2023-08-09
>
> Thank you for providing valuable feedback on our paper. We will explain each point one by one. If you have new questions or ideas, please don't hesitate to let us know.
>
> 1. Statement for the smoothing term $d(\mu)=\lambda/2||\mu||^2$
>
> We appreciate your inquiry regarding the smoothing term, as it's an excellent question that we'll delve into here in detail. Firstly, to address the computational challenges stemming from the non-differentiability of the hinge loss, we employ a smoothing approach by subtracting a prox function $d(\mu)$. This smoothing technique not only simplifies computations but also introduces more convenient stopping criteria. From the viewpoint of the purpose, the incorporation of the smoothing term $d(\mu) = \lambda/2|||\mu||^2$ is primarily for computational ease.
>
> Secondly, the prox parameter $\gamma$ we've chosen for specific computations is extremely small ($\gamma\le1/n$), and the vast majority of $\mu_i, i\in[n]$, remain at $0$ during iterations. As a result, the smoothing term has minimal impact.
>
> Lastly, even if we were to optimize the original hinge loss directly, the gradient descent algorithm would still instigate implicit regularization. This is because implicit regularization fundamentally arises from the algorithm itself, rather than from the process of smoothing. However, direct optimization of the original hinge loss poses considerable theoretical challenges (owing to non-differentiability). Therefore, the introduction of smoothness is a clever approach, significantly benefiting both computational and theoretical aspects.
>
> 2. Assumption 1 holds on real data？
>
> Assumption 1, in reality, doesn't overly constrict the analysis of real-world data. As we stress in the paper, while our proof relies on Assumption 1, the experimental outcomes present compelling evidence that its strict adherence isn't crucial for the success of our approach. To exemplify, we generate data from the $t$-distribution, which clearly doesn't meet Assumption 1, yet the experimental results remain surprisingly strong. This showcases the potential for relaxing the constraints of Assumption 1 in practical scenarios.
>
> To provide a more tangible illustration, we've incorporated a practical example. In this case, we applied our algorithm to the colon-cancer dataset sourced from the LIBSVM website (https://www.csie.ntu.edu.tw/~cjlin/libsvmtools/datasets/). This dataset comprises 62 samples with 2000 features—a quintessential high-dimensional sparse classification challenge. With 42 samples as the training set, 10 as the testing set, and 10 for validation, our approach achieves impressive results. Specifically, the final prediction accuracy on the training set is 0.8, sensitivity reaches 0.75, and specificity stands at 0.833—an exceptional performance. Looking ahead, our plans encompass introducing new real data analysis results in the revised version.
>
> 3. Discussion of the empirical effectiveness of the proposed algorithm
>
> SVM finds extensive use in classification tasks such as face detection, text categorization, image classification, bioinformatics, handwriting recognition, and medical data analysis. Yet, when confronted with high-dimensional data, SVM can encounter challenges, especially when the sample count is significantly smaller than the feature count. Take, for instance, the colon-cancer dataset we previously examined, where features reach 2000, while samples total only 62. Similar situations arise in text and image categorization, where image dimensions can be notably high. However, it's essential to acknowledge that high dimensionality often accompanies sparsity. Consider the news20 dataset from the libsvm website—featuring 19,996 samples and a staggering 1,355,191 features, yet maintaining a sparse 0.034% density. In such scenarios, conventional SVM algorithms can struggle to yield precise classifications. This is where our proposed algorithm, facilitated by implicit regularization, steps in.
>
>  As demonstrated in our earlier analysis of the colon-cancer data, our algorithm thrives in the realm of high-dimensional and sparse data. It accomplishes effective classification outcomes while ensuring computational simplicity. In essence, our proposed algorithm holds immense promise in addressing classification challenges within today's high-dimensional, sparse data landscapes. Its potential impact spans across practical applications like face detection, text and image categorization, and medical data analysis. This algorithm has the potential to tackle intricate classification tasks encountered in real-world settings, providing viable solutions.
>
> Furthermore, our algorithm excels in terms of algorithmic effectiveness. In the default experimental setup ($m=10$), the average runtime for executing the gradient descent algorithm is 2.75 seconds. This represents an 18% improvement in speed compared to the Lasso algorithm.

---

### Official Review · Reviewer_6PiA · 2023-07-06

**Soundness:** 3 good
**Presentation:** 2 fair
**Contribution:** 3 good
**Rating:** 6
**Confidence:** 5

**Summary:**

The paper proposes an iterative/implicit regularization algorithm for sparse SVM, using an Hadamard product overparametrization of the iterate $\beta = u \odot u - v \odot v$. In addition, a Nesterov smoothing of the Hinge loss (replaced by its Moreau envelope, a term that is lacking in the paper) is performed.

The main result is Theorem 2, stating that results equivalent to that of explicit penalization approach can be obtained, under similar assumptions (subgaussianity of the samples, low incoherence of the design matrix), in terms of oracle error with high probability.

**Strengths:**

Implicit regularization is a very active area of research; implicitly regularized algorithms usually require a lower computational budget. Extending known results from regression and sparse matrix factorization to classification is interesting.

**Weaknesses:**

- no code is included in the supplementary material, and no code release is mentioned in the paper.
- the writing of the paper can be strongly improved; it would benefit from being proofread by a proficient English reader.
    - E.g. in the first two lines, do not use "the" in "based on the gradient-based methods", and use plural in "models, such as the deep learning model" (there are more than one DL model...)
    - L20, "tend to converge to the global minimum." : a global minimum
    - "$s$ is number of signals"
    -  near-oracle rate is achievable via explicit regularization using explicit regularization
    -  show that gradient descent estimator: missing "the"
    - L110 why "that"?
    - L120 repeats the same idea twice
    - proxy parameter instead of prox parameter, and it would be clearer to call it "smoothing paramter" or Moreau envelope paramter.
    - etc etc.
- the paper also lacks rigor and clarity in its writing in many places, e.g. "the zero component is initialized close to zero". What the authors mean is that the coordinates which are zero **in $\beta^*$** are initialized closed to 0 **in the optimization variable**. $\beta^*$ is not even defined.
- the algorithm's stopping criterion is based on $\mu_i$, and the algorithm stops
when all $\mu_i$'s are negative. This means that all training points are classified in their observed classes. In the case of label noise, doesn't this lead to overfitting? How is this stopping criterion related to the bound on number of iterations $t$ in Theorem 1 and Proposition 2?
- why is the performance measure not $\Vert \beta - \beta^* \Vert$ in the experiments? Why are the vectors normalized? This does not match the metric used in the theoretical results.
- the approach simply drops the two L2 regularization terms, $\Vert w \Vert^2 + \Vert v \Vert^2$, that come from the original L1 regularization term $\Vert \beta \Vert_1$. Why?
- the authors claim that there is no parameter in their approach, but the smoothing of the hinge loss does require one.
- why is $d_\gamma$ called a prox-function, when it's just a strongly convex function? why introducing a general notation when only the squared L2 norm is used (l142)
- why are there 3 orange curves in Fig 1c? shouldn't those be shaded plots? Where is the rud curve? can you sort the legend by increasing value instead of having 1e-10, 1e-4, 1e-6 in that order?
- How is Lasso's (L1 regularized SVM) regularization parameter tuned?
- Unless I'm mistaken, thee iterates of the proposed method are never exactly zero. Some thresholding needs to be performed in order to avoid dense vectors and many false positive (Fig 2c for example). How is this done?


- "we present finite sample performances of our method in comparisons with the Lasso estimator": SVM is for classification, Lasso is for regression. How can you compare them? Do you mean L1 regularized SVM?
- under the assumption of Sub-Gaussian distribution: mention that this is about the noise
- L114 the fact that there is a generative model with "true parameters" is completely missing
- L139 it's a saddle point problem not a saddle point function
- In Thm 1 there is no need to mention randomness, as the generative model does not come into play. Making this a theorem is a bold move.
- the legends in the plots are too small to be read.

### References
- The proposed Hadamard parametrization is, up to my knowledge, proposed by Vaskevicius. This could be explicitely stated L43 (though it is currently mentioned, it comes later in the paper)
- A generic algorithm for any type of sparse regularization is proposed in "Iterative regularization for low complexity regularizers" (Molinari et al, 2022). In particular, handling of sparse classification is proposed in example 1. How does the proposed approach compare?
- the authors should cite "Smoothing and first order methods: a unified framework" by Beck and Teboulle (2012). Their thm 1 is a known result of this paper, which holds for "inf-convolution"/Nesterov smoothing regardless of the function being smoothed.

**Questions:**

see weaknesses above

**Limitations:**

no societal impact

---

> ### Author Rebuttal · Authors · 2023-08-09
>
> Thank you for carefully reading our paper and sharing valuable feedback. Due to character limits, we can't address every question separately. We will fix all writing mistakes and unclear parts you've pointed out in the revised version. For other questions, we address them here. If you have more questions, please let us know without hesitation.
>
> - Code.
>
> In our revised version, we will add a GitHub link. Notably, our algorithm is straightforward. We obtain the lasso estimator through R packages 'sparseSVM' or 'penalizedSVM.' The oracle estimator can be derived from standard SVM.
>
> - Supplement to the setting of initialization and true parameters $\beta^*$.
>
> We explain more about initial setting. Ideally, we'd initialize $w$ and $v$ to mirror the sparsity pattern of $\beta^*$—meaning $w_S$ and $v_S$ would be non-zero, while the values outside the support would be set to 0. Unfortunately, this isn't feasible due to the unknown support of $\beta^*$. Instead, we initialize as $w_0=v_0=\alpha1_{p\times 1}$. This approach strikes a balance: zero components receive almost-zero initial values ($\alpha$ being very small), and nonzero components receive non-zero initializations. Regarding the definition of real $s$-sparse parameter $\beta^*$, which we forgot to declare in the main text, $\beta^*=\arg\min_{\beta}{\mathbb E}(1-yx^T\beta)_+$, we will also add it in the revised version.
>
> - Discussion of the stopping criterion.
>
> In an ideal setup, we can rely on $\mu_i=0$ to stop iterations, such as in the simulations. In general, we still keep the maximum number of iterations $T_1$ as a stopping condition, (see Algorithm 1), and this stopping condition can be applied to all cases. If we know all parameters, we can estimate the ideal iteration range and subsequently set $T_1$.
>
> - why is the measure normalized?
>
> We used normalized metrics to compare with the oracle estimator. In simulations, the oracle estimator exhibits a large error before normalization, and a smaller error after normalization. For instance, when dealing with signals like (5,6,7,8) (normalized to (0.38,0.45,0.53,0.61)), the oracle estimator produces values of (1.27,1.57,1.88,2.11) (normalized to (0.37,0.45,0.54,0.61)). The GD estimator yields (3.3,3.97,5.01,5.74) (normalized to (0.36,0.43,0.54,0.62)).
>
> While we used the normalized metric for comparison, note that the GD estimator performs well under the $||\beta_t-\beta^*||$ metric.
>
> - Reason why we drop the two L2 regularization terms.
>
> We explain more here. First, the motivation of  implicit regularization comes from some interesting phenomena in deep learning. Applying deep learning to regression and classification, the regression function or classifier is represented by a deep neural network. But the loss function is nonconvex. In addition, neural networks are over-parameterized, which makes the regression or classification ill-posed statistically. However, people found that simple algorithms such as gradient descent tend to find the global minimum of the loss function.
>
> To understand why, [B,C] suggest that generalization stems from implicit regularization of optimization algorithms. They observed that in over-parameterized models , the algorithm, usually a variant of the gradient descent, prefers solutions that generalize well. Without adding any regularization term, it is the implicit preference of the algorithm that acts as a regularizer. For this reason, in recent work on implicit regularization in statistical models (e.g., regression), scholars follow this line by not adding any regularization terms and focus on the implicit regularization of the algorithm.
>
> To be specific, in line 122, the new optimization is $\min{\cal E}_{{\cal Z}^n}(a,c)+\lambda(||a||^2+||c||^2)$. Following neural network training, we remove $\ell_2$ norm, a practice that is common in implicit regularization work[10, 28, 34].
>
> [B] Neyshabur, B., Tomioka, R. and Srebro, N. (2015). In search of the real inductive bias: On the role of implicit regularization in deep learning.
>
> [C] Zhang, C., Bengio, S., Hardt, M., Recht, B. and Vinyals, O. (2017). Understanding deep learning requires rethinking generalization.
>
> - The smoothing requires one parameter.
>
> The smoothing does require a parameter $\gamma$, but the constraint on $\gamma$ is quite relaxed, requiring no complex tuning.
>
> - why is $d(\gamma)$ called a prox-function?
>
> Our smoothing follows [22], hence certain terms are adopted from the original paper. We'll modify the expressions in the revised version.
>
> - Explanation of Fig 1b and Fig 1c.
>
> Fig 1b shows strong signal magnitudes over iterations, while Fig 1c shows error term magnitudes over iterations. Mean values are presented in both figures, omitting quartile-shaded areas for clarity.
>
> Only 3 orange curves in Fig 1c results from larger estimation error magnitude for 1e-4 compared to the other initial values. For enhanced clarity, an "Error Components" Fig is added in "General.pdf". This figure aligns with Proposition 1: $\Arrowvert\beta_{t}\odot1_{S_1^c}\Arrowvert_{\infty}=\Arrowvert w_{t}\odot w_t\odot1_{S_1^c}-v_{t}\odot v_t\odot 1_{S_1^c}\Arrowvert_{\infty}\lesssim (\sqrt{\alpha})^2=\alpha$.
>
> We'll reorganize the legend in ascending order based on values, as you suggested.
>
> - Tuning in Lasso
>
> The tuning methods in ‘sparseSVM’ and ‘penalizedSVM’  are cross-validation by default.
>
> - Thresholding procedure.
>
> After getting GD estimator and lasso estimator, we choose a threshold (like 1e-5), coordinates less than the threshold are counted as 0.
>
> - Discussion of References.
>
> Comparing with (Molinari et al., 2022), we have these distinctions:
>
> 1. Our focus is on gradient descent without explicit regularization, while Molinari works on primal-dual algorithm with regularization.
> 2. We analyse error bound of iterates, while Molinari analyse stability bound.
> 3. We introduce reparametrization to classification, studying gradient descent dynamics. This novel aspect is unexplored in classification.

---

> > ### Comment · Reviewer_6PiA · 2023-08-16
> >
> > I thank the authors for their answer, that only answer my questions partially (in particular, if normalization strongly affects the results, this should be mentioned and both settings included in the paper)
> >
> > Reading the paper again, I found more issues:
> > - L122 it is claimed that one can minimize $\mathcal{E}(a, b)$, but that is definitely not clear: the cosntraint $a \odot c = \beta$ is missing, and plugging $a$ and $b$ in \$mathcal{E}$ yields a $a \odot a - c \odot c$ which is definitely not equal to $\beta$. This whole paragraph is unclear.
> > - L491 has "based on the definition of $T^*$, however $T^*$ in Proposition 1 is defined only with a $\mathcal{O}$ formulation, not a precise value. line 463, the same issue appears. Also the last $(ii)$ above the inequalities L498 should be a $(iii)$.
> > - L497 there is a ':' before the $>$. Overall this gives the very strong feeling that the paper's writing was rushed and that more concerning issues may have slipped through.
> > - As the code was not submitted, it was impossible to reproduce the numerical experiments
> > - Highlighting the sensitivity of the method with respect to $\gamma$ also seems mandatory for acceptance in my opinion
> >
> > Overall, I am not completely convinced by the author's answer (for example, their definition of $\beta^*$ as $\mathrm{argmin} \mathbb{E} (1 - Y X^\top \beta)_+$ does not match the experiments, where $y_i = \mathrm{Bernoulli}(x_i^\top \beta^*)$ is used; what is the guarantee that $\beta^*$ is effectively the minimizer of the population hinge loss? The Bernoulli distribution rather hints at a logistic loss)
> >
> > In my opinion, there are too many issues in the current version of the paper to accept it without a thorough second round of reviewing. I keep my vote for rejection.

---

> > > ### Author Response · Authors · 2023-08-17
> > > **Detailed Replies for Your Comments (Part 1/3)**
> > >
> > > Thank you very much for your valuable comments and suggestions.
> > >
> > >  **I thank the authors for their answer, that only answer my questions partially (in particular, if normalization strongly affects the results, this should be mentioned and both settings included in the paper).**
> > >
> > > Thank you very much for your comments.  We sincerely apologize for not answering your questions one-by-one, which is indeed due to the required character limits. Following your suggestion, we will provide detailed explanations on the normalization in Section 4 and add the numerical results of all the  methods without normalization in the supplemental file to provide a comprehensive comparison.
> > >
> > > **Q1: L122 it is claimed that one can minimize ${\cal E}(a, b)$, but that is definitely not clear: the cosntraint $a\odot b=\beta$ is missing, and plugging $a$ and $b$ in ${\cal E}$ yields a $a \odot a -c \odot c$ which is definitely not equal to $\beta$. This whole paragraph is unclear.**
> > >
> > > Thanks a lot for your comments. We will rewrite the corresponding part, including the statement on the contraint $\beta=a\odot c$ in Line 122, and provide more detailed explanations on this issue in the revised version. Specifically,  by Lemma 1 in [1], there holds that
> > >
> > > $\inf_{\beta}\frac1n\sum_{i=1}^n(1-y_ix_i^T\beta)_++\lambda||\beta||_1$ equals to
> > >
> > > $\inf_{a,c}\frac1n\sum_{i=1}^n(1-y_ix_i^Ta\odot c)_++\lambda(||a||^2+||c||^2)/2$
> > >
> > > and then the right-hand side can be further reparameterized by using  $w=\frac{a+c}{2}$ and $v=\frac{a-c}{2}$. Clearly, we have $\beta=w\odot w-v\odot v$ and $2p$ new parameters $w$ and $v$ are introduced to over-parameterize the original optimization problem. Finally, we drop the explicit penalty and focus on the empirical  $\frac1n\sum_{i=1}^n(1-y_ ix_i^T(w\odot w-v\odot v))_+$.  It is worthynoting that the  same treatment is commonly used in the literature of implicit regularization [2,3,4].
> > >
> > > [1] Hoff, P. D. (2017). Lasso, fractional norm and structured sparse estimation using a Hadamard product parametrization.
> > >
> > > [2] Vaskevicius, T., Kanade, V., & Rebeschini, P. (2019). Implicit regularization for optimal sparse recovery.
> > >
> > > [3] Fan, J., Yang, Z., & Yu, M. (2022). Understanding implicit regularization in over-parameterized single index model.
> > >
> > > [4] Zhao, P., Yang, Y., & He, Q. C. (2022). High-dimensional linear regression via implicit regularization.
> > >
> > > **Q2: L491 has "based on the definition of T, however T in Proposition 1 is defined only with a ${\cal O}$ formulation, not a precise value. line 463, the same issue appears. Also the last $(ii)$ above the inequalities L498 should be a $(iii)$.**
> > >
> > > Thank you very much for your comments.  We apologize for missing the exact form of $T^*$ in the text. Actually, $T^*$ is defined as $T^*=\log(1/\alpha)/(4\sigma\eta\log n)$, and thus $a_1$ in line 463 is $1/(4\sigma)$. Moreover,  the inequality notations above L498 are also fixed in the revised version.
> > >
> > > **Q3: L497 there is a $':'$ before the $>$. Overall this gives the very strong feeling that the paper's writing was rushed and that more concerning issues may have slipped through.**
> > >
> > >
> > > We sincerely apologize  again for the typos and unclear expressions in this paper. We will double check this paper to correct the typos and also find some proficient English readers to proofread it.
> > >
> > > **Q4: As the code was not submitted, it was impossible to reproduce the numerical experiments.**
> > >
> > > Thanks a lot for your comment. For your convenience,  we upload the code for Algorithm 1 in the following link.(https://drive.google.com/file/d/1ZyMz1KMaI5tR9TeAUMFHWXBuKfzWRiQ9/view?usp=sharing)  Note that this link and the shared file do not contain any relevant author information.

---

> > > ### Author Response · Authors · 2023-08-17
> > > **Detailed Replies for Your Comments (Part 2/3)**
> > >
> > > **Q5: Highlighting the sensitivity of the method with respect to $\gamma$ also seems mandatory for acceptance in my opinion.**
> > >
> > > Thank you very much for your suggestion.  We will add the sensitivity analysis about $\gamma$ in the revision. Specifically, the detailed experimental setup follows Lines 250-261 and $\gamma$ takes the value in $[2.5\times10^{-5},1\times 10^{-3}]$. The experiments are replicated for multiple times, and the averaged numerical results in terms of  estimated signal strengths are shown in the following table, where the standard deviation is in parentheses.
> > >
> > > | $\gamma$                                  | $2.5\times10^{-5}$ | $5\times10^{-5}$ | $7.5\times10^{-5}$ | $1\times10^{-4}$ |
> > > | ----------------------------------------- | ------------------ | ---------------- | ------------------ | ---------------- |
> > > | $\|\|\beta_t-\beta^*\|\|/\|\|\beta^*\|\|$ | 0.480(0.053)       | 0.469(0.054)     | 0.463(0.055)       | 0.460(0.057)     |
> > > | signal 1                                  | 5.316(0.482)       | 5.427(0.522)     | 5.471(0.540)       | 5.503(0.534)     |
> > > | signal 2                                  | 5.340(0.626)       | 5.444(0.642)     | 5.508(0.648)       | 5.533(0.675)     |
> > > | signal 3                                  | 5.036(0.813)       | 5.165(0.844)     | 5.235(0.867)       | 5.253(0.876)     |
> > > | signal 4                                  | 5.350(0.653)       | 5.481(0.653)     | 5.548(0.665)       | 5.583(0.697)     |
> > > | $\gamma$                                  | $2.5\times10^{-4}$ | $5\times10^{-4}$ | $7.5\times10^{-3}$ | $1\times10^{-3}$ |
> > > | $\|\|\beta_t-\beta^*\|\|/\|\|\beta^*\|\|$ | 0.455(0.058)       | 0.461(0.058)     | 0.470(0.057)       | 0.479(0.055)     |
> > > | signal 1                                  | 5.557(0.578)       | 5.491(0.596)     | 5.385(0.586)       | 5.289(0.574)     |
> > > | signal 2                                  | 5.595(0.680)       | 5.543(0.664)     | 5.438(0.658)       | 5.348(0.652)     |
> > > | signal 3                                  | 5.299(0.869)       | 5.238(0.856)     | 5.150(0.830)       | 5.065(0.812)     |
> > > | signal 4                                  | 5.651(0.671)       | 5.573(0.670)     | 5.484(0.650)       | 5.381(0.627)     |
> > >
> > > From the table, it is clear that the choice of $\gamma$ is less sensitive in the sense that the estimation error and the estimated strengths of the signals are very close to each other, and the estimation accuracy slightly decreases when $\gamma$  increases, which is still within the acceptable range.

---

> > > ### Author Response · Authors · 2023-08-17
> > > **Detailed Replies for Your Comments (Part 3/3)**
> > >
> > > **Overall, I am not completely convinced by the author's answer (for example, their definition of as does not match the experiments, where is used; what is the guarantee that is effectively the minimizer of the population hinge loss? The Bernoulli distribution rather hints at a logistic loss)**
> > >
> > > Thank you very much for your comments. In fact, the generating scheme of $y$ adopted in the original submission follows the similar treatment as in [5,8], where the task of variable selection for support vector machine is considered.  In the revision, we also use some other generating schemes, which is adopted in [6,7]. Specifically, we generate random data from the following two models.
> > >
> > > - Model 1: $X\sim MN(0_p,\Sigma)$, $\Sigma=(\sigma_{ij})$ with nonzero elements $\sigma_{ij}=0.4^{|i-j|}$ for $1\le i,j\le p$, $P(y=1|X)=\Phi(X^T\beta^*)$, where $\Phi(\cdot)$ is the cumulative density function of the standard normal distribution, $\beta^*=(1.1,1.1,1.1,1.1,0,\ldots,0)^T$ and $s=4$.
> > > - Model 2: $P(Y=1)=P(Y=-1)=0.5$, $X|(Y=1)\sim MN(\mu,\Sigma)$, $X|(Y=-1)\sim MN(-\mu,\Sigma)$, $s=5$, $\mu=(0.1,0.2,0.3,0.4,0.5,0,\ldots,0)^T$, $\Sigma=(\sigma_{ij})$ with diagonal entries equal to 1, nonzero entries $\sigma_{ij}=-0.2$ for $1\le i\not=j\le s$ and other entries equal to $0$. The bayes rule is $sign(1.39X_1+1.47X_2+1.56X_3+1.65X_4+1.74X_5)$ with bayes error $6.3$%.
> > >
> > > The setup of parameters follows Lines 250-261 and  the experiments are replicated for multiple times. The averaged estimation and prediction results are shown in the following table.
> > >
> > > | Generating Model                                             | Model 1      | Model 2      |
> > > | ------------------------------------------------------------ | ------------ | ------------ |
> > > | $\|\|\beta_t/\|\|\beta_t\|\|-\beta^*/\|\|\beta^*\|\|\|\|$(GD) | 0.164(0.086) | 0.150(0.106) |
> > > | $\|\|\beta_t/\|\|\beta_t\|\|-\beta^*/\|\|\beta^*\|\|\|\|$(Oracle) | 0.155(0.076) | 0.105(0.048) |
> > >
> > >
> > > From this table, we can easily see that the GD estimator can approach the oracle estimator in terms of estimation error. Note that  the full numerical results of the newly added examples are added at  very beginning of supplemental file, which further indicate the performance of the proposed method.
> > >
> > > [5] Zhang, H. (2006). Variable  Selection  for Support  Vector Machines via Smoothing  Spline  ANOVA.
> > >
> > > [6] Peng, B., Wang, L., & Wu, Y. (2016). An error bound for l1-norm support vector machine coefficients in ultra-high dimension.
> > >
> > > [7] Zhang, X., Wu, Y., Wang, L., & Li, R. (2016). Variable selection for support vector machines in moderately high dimensions.
> > >
> > > [6] He, H., Lv, S. \& Wang, J. (2020). Variable Selection  for Classification  with Derivative-induced regularization.
> > >
> > > **In my opinion, there are too many issues in the current version of the paper to accept it without a thorough second round of reviewing. I keep my vote for rejection.**
> > >
> > > We deeply apologize for not answering your questions one-by-one in our previous response. To be honest, our original response contains more than 12000 characters, however, the character limit is up to 6000 in this year. We really appreciate your precious comments and suggestions on this paper, and will revise the paper correspondingly, including proofread by some proficient English reader, enlarge the legends in the figures and so on.
> > >
> > > To the end, we want to thank you again and hope you can reconsider this paper.

---

> > > ### Author Response · Authors · 2023-08-17
> > > **Supplementary answers for some of your first-round questions.**
> > >
> > > We sincerely apologize for not answering your questions one-by-one, which is indeed due to the required character limits. We try the best to answer the rest of your previous questions below.
> > >
> > > **Why we design initialization this way.**
> > >
> > > Thanks a lot for your comment. In this paper, we initialize $w_0$ and $v_0$ as $w_0=v_0=\alpha1_{p\times 1}$, and such a construction provides a good compromise. Specifically, the zero components get nearly zero initializations, which are the majority under the sparsity assumption, and nonzero components get nonzero initializations. Even if we initialise each component at the same value, the non-zero components move quickly, while the zero components remain small. This is how overparameterization differentiate active components from
> > > inactive components, and similar treatment is widely adopted in literature [2,3,4]. We can see this phenomena in Figures 1 and  6.
> > >
> > >  **Why we drop $\ell_2$ norm.**
> > >
> > > Thanks a lot for your comment. We want to clarify that the research goal of this paper is to explore the phenomenon of implicit regularization. This phenomenon was first observed in deep learning, in applied neural networks, gradient descent and its variants tend to find the global minimum despite overparameterization. It is worthnoting that  the regularization term is often not added in neural network training. [9,10] show that it is the implicit preference of the optimization algorithm itself will play a role of regularization.
> > >
> > > Motivated by this phenomenon, research on implicit regularization in other learning tasks has gradually increased in recent years. [2,3,4] usually follow this line of thought by not adding any explicit regularization term to the optimization objective and focusing on the role of the optimization algorithm itself. We also follow this line to explore the implicit preference of gradient descent in SVM, both theoretically and empirically.
> > >
> > > **Unclear expressions and unreadable images.**
> > >
> > > Thank you for your comments, we will revise these issues according to your comments one by one, for example, declaring that "Lasso estimator" specifically refers to $\ell_1$-regularized SVM; changing "saddle point function " to "saddle point problem", and we will make the figures more readable in the revised version for readers' convenience.
> > >
> > > [9] Neyshabur, B., Tomioka, R. and Srebro, N. (2015). In search of the real inductive bias: On the role of implicit regularization in deep learning.
> > >
> > > [10] Zhang, C., Bengio, S., Hardt, M., Recht, B. and Vinyals, O. (2017). Understanding deep learning requires rethinking generalization.

---

> > > > ### Comment · Reviewer_6PiA · 2023-08-21
> > > >
> > > > This second round of answer lifts some doubts that I had. I have increased my score.

---

> > > > > ### Author Response · Authors · 2023-08-21
> > > > > **Reply to the Comment**
> > > > >
> > > > > Thank you very much for your valuable insights. They have allowed us to gain a deeper understanding of the proposed method and have greatly improved the quality of this paper. We will make revisions to the paper and proofread it in accordance with your suggestions.

---

### Official Review · Reviewer_S4hZ · 2023-07-06

**Soundness:** 4 excellent
**Presentation:** 3 good
**Contribution:** 3 good
**Rating:** 6
**Confidence:** 2

**Summary:**

This paper design a regularization-free algorithm for high-dimensional support vector machines (SVMs) by integrating over-parameterization with Nesterov's smoothing method, and provide theoretical guarantees for the induced implicit regularization phenomenon.

**Strengths:**

This paper provides a regularization-free gradient method that has proven to be effective in practice and is supported by some theoretical guarantees. The theoretical results are novel, and the experiments are comprehensive.

**Weaknesses:**

The theoretical results are relatively weak.

1. The constants $c1\sim c{4}$ in Theorem 2 are not clearly specified, so we cannot see the trend of these constants changing with the problem size (are they independent of $p$?), which weakens the results.

2. The author did not compare the strength of the $\delta$-incoherence assumption and the RIP (Restricted Isometry Property) assumption (only a comparison for verifying difficulty was made). This comparison should be included.

3.The restrictions on the initial values in Assumption 2 may seem somewhat stringent. The author may provide an explanation for why they are designed this way.

**Questions:**

Please refer to the Weakness.

---

> ### Author Rebuttal · Authors · 2023-08-09
>
> We appreciate the time and effort you put into reviewing our work. We've reviewed your comments and summarized our responses below. Please let us know if you have any additional comments or concerns.
>
> 1. Clear specifications about the constants $c_1\sim c_4$ in Theorem 2.
>
> Thank you for highlighting this for us. We prepare a more detailed explanation of the constants. In Theorem 1, the constants $c_1$ and $c_2$ arise from inequalities related to sub-gaussian random variables, these constants remain unrelated to $n$ or $p$. Likewise, the constants $c_3$ and $c_4$ are associated with the interval of iterations, and we can give the specific form of this interval, $[5\log(m/\alpha^2)/\eta m,\log(1/\alpha)/(4\sigma\eta\log n)]$, thereby setting $c_3=5$ and $c_4=1/(4\sigma)$. Notably, our assumptions do not involve $\sigma$, so $c_3$ and $c_4$ remain independent of $n$ or $p$. In conclusion, these constants do not undermine the theoretical results.
>
> 2. Comparison of the $\delta$-incoherence assumption and the RIP assumption.
>
> This is a pertinent question, and we appreciate the chance to provide a clearer explanation of these two assumptions. The incoherence and RIP are powerful performance measures for guaranteeing sparse recovery and have been widely used in many contexts. Sub-Gaussian matrices satisfy low-incoherence and RIP property with high probability. In fact, incoherence and RIP imply each other[E], showing their close connection.
>
> However, compared to the incoherence assumption, the RIP poses a couple of challenges. Firstly, RIP verification becomes NP-hard when dealing with pre-constructed design matrices. Secondly, the RIP's formal complexity renders it challenging to compute. In prior explorations of implicit regularization in matrix factorization and linear regression, some scholars have flagged issues with RIP assumptions. For instance, RIP cannot capture the behaviour of $\ell_2$-loss in the high noise regime [M], and RIP condition could potentially be replaced [28, 34]. Consequently, we opt for the more flexible incoherence assumption in this paper, a choice supported by recent work [18].
>
> [E] Candes, E.J., 2008. The restricted isometry property and its implications for compressed sensing. *Comptes rendus. Mathematique*, *346*(9-10), pp.589-592.
>
> [M] Jianhao M and Fattahi S. Global convergence of sub-gradient method for robust matrix recovery: Small initialization, noisy measurements, and over-parameterization. *Journal of Machine Learning Research* 24.96 (2023): 1-84.
>
> 3. Detailed explanations about Assumption 2.
>
> The details in Assumption 2 are also worth looking into. In addition to explanations in the paper, we're providing more insights here based on both theory and experiments. Firstly, the assumptions about the starting value $\alpha$, parameter $\gamma$, and step size $\eta$ mainly stem from the theoretical side of the algorithm. For instance, $\alpha$ controls the strength of the estimated weak signals and error components, $\gamma$ manages the approximation error in smoothing, and $\eta$ affects how accurately we estimate strong signals. So, assumptions about these are necessary. Now, for the assumptions about $\alpha$ and $\eta$—where we say $\alpha \lesssim 1/p$ and $\eta \lesssim 1/(\kappa \log p)$—we're not sticking strictly to the $\le$ sign, but $\lesssim$. These assumptions aren't too strict, and in our tests, making them really small isn't necessary for good results. For example, we tried $\eta = 0.5$ (much larger than $1/(\kappa \log p)$), and even with a larger $\alpha$ like $10^{-4}$, we still got good estimates.
>
> Speaking about $\gamma$, we're strict in the assumption, but in our experiments, we can be a bit flexible. We tested different $\gamma$ values, like $10^{-4}$, $10^{-3}$, and $5 \times 10^{-3}$, for estimation and finally we got the estimation errors of $0.383$, $0.404$ and $0.519$, and the four strong signals are estimated as $[5.8,5.8,6.6,6.6]$, $[5.6,5.6,6.4,6.4]$ and $[4.4,4.5,5.2,5.2]$, respectively. Even with larger $\gamma$ values—like scaling it up a lot—we still got good results in estimating errors and strong signals. To sum it up, the setting of parameters is very common. In related studies like [10,18, 28, 34], they make certain assumptions about initial values and step sizes. In experiments, we can relax the rules a bit. There's no need to make the parameters super small to get the results we want.

---

> > ### Comment · Reviewer_S4hZ · 2023-08-12
> >
> > Based on the author's response, I intend to raise my score to 6.

---

> > > ### Author Response · Authors · 2023-08-12
> > > **Reply to the Comment**
> > >
> > > Thank you so much for your help. Your comments help us improve and gain a better understanding of the method. We appreciate your time and effort in reviewing our work.

---

### Official Review · Reviewer_Z81p · 2023-07-13

**Soundness:** 4 excellent
**Presentation:** 4 excellent
**Contribution:** 3 good
**Rating:** 8
**Confidence:** 3

**Summary:**

The paper tackles the problem of implicit regularization for classification in the context of over-parameterization. Starting from a $L^1$ regularized SVM, they voluntarily over-parameterize the feature vector $\beta = w \odot w - v \odot v $ using two vectors $w, v \in \mathbb{R}^p$.

The optimization process then consists in (i) dropping the explicit regularization, (ii) smoothing the hinge loss function, (iii) applying gradient descent with early stopping to get the solution.

A theoretical analysis of the estimated parameter is provided, showing under mild conditions that the proposed scheme behaves as well as the explicit regularization scheme if the algorithm is stopped at a certain time.

Numerical experiments highlight the benefits of the approach on synthetic data.

**Strengths:**

- The topic is of interest for the machine learning community.
- The paper is well written and mathematically sound.
- The developed method is novel.
- The experimental study is convincing.

**Weaknesses:**

- No real weakness

**Questions:**

- The problem considered here is a linear SVM with $L^1$ penalization. Would the results presented here still hold for a non-linear SVM based on a finite-dimensional feature map ?

- At one point, you drop the explicit regularization to only minimize an empirical risk. What is the ingredient in the algorithm that allows to recover properties similar to the explicit regularization ? Once the regularization is dropped, we could say that the original problem considered another one - e.g. Tychonov regularization - but no the solution presented here retains specific properties usually associated with the $L^1$ penalty. Could you comment on that ?

Typos/remarks:

- Line 18: on gradient based methods
- I believe it would be of interest to cite [A] alongside [1] and [27] in the introduction, given the link they explore with the Lasso problem.
- Line 44: Why $u_0$ and not $w_0$ ? Please be consistent with the notation from line 42
- Line 47: "near-oracle rate is achievable via explicit regularization using explicit regularization"
- Line 145: "and has an explicit form that" writes ?
- Line 146: given that you already use a superscript in equation (4) for $\mathcal{Z}^n$, you should use a subscript here to avoid overcharge.
- Line 229: "that scales as $\mathcal{O}$"
- Line 236: the subscripts are not correct (two times the complement)

[A]: Iterative regularization for convex regularizers; Molinari, Cesare and Massias, Mathurin and Rosasco, Lorenzo and Villa, Silvia; AISTATS 2021

**Limitations:**

Limitations have overall been addressed.

---

> ### Author Rebuttal · Authors · 2023-08-09
>
> We appreciate the valuable feedback you've provided on our paper. We will go through each point and provide explanations. If you have any new questions or ideas, please feel free to share them with us.
>
> 1. Future work about implicit regularization in Non-linear SVM.
>
> Exploring implicit regularization in nonlinear SVM is a direction we're considering for future work.
> Note that the proposed method can be extended to the non-linear SVM with a finite-dimensional feature map, which includes the polynomial kernel. Specifically, it is clear that the kernel SVM with linear kernel is exactly the same as our proposed method. Once the polynomical kernel is used, we may reparameterized the corresponding coefficent with some high order vectors, and the theoretical results may be established with some modificatiokns. In the revised version, we have added some detailed discussions on the potential route of extending the proposed method to the nonlinear SVM with finite-dimensional feature map.
>
> 2.  What allows our algorithm to recover properties similar to the explicit regularization？
>
> Implicit regularization is the focus of our paper, and we provide a comprehensive explanation here. Many optimization challenges in networks involve aspects like nonconvexity [C1], nonlinearity, and over-parameterization. These problems potentially lead to subpar performance in regression or classification tasks from a statistical standpoint.
>
> However, what's interesting is that practical observations show that simple algorithms like (stochastic) gradient descent often manage to find the global minimum of the loss function, even when faced with nonconvexity. This happens without the need for explicit regularization [B, C2]. In other words,  in over-parametrized statistical models (like our experimental setup), although the optimization problems consist of bad local minima with large generalization error, the choice of optimization algorithm, usually a variant of gradient descent algorithm, usually guard the iterates from bad local minima and prefers the solution that generalizes well. As a result, we don't introduce an extra regularization term to the optimization goal. Instead, the optimization algorithm itself showcases implicit preferences that effectively play the role of regularization.
>
> We can interpret the iteration count $t$ as a kind of regularization parameter, similar to $\lambda$ in the lasso penalization algorithm. Our approach attains the desired performance when the number of iterations falls within the appropriate range.
>
> [C1]Yun, C., Sra, S. and Jadbabaie, A. (2019). Small nonlinearities in activation functions create bad local minima in neural networks. In *International Conference on Learning Representations*.
>
> [B] Neyshabur, B., Tomioka, R. and Srebro, N. (2015). In search of the real inductive bias: On the role of implicit regularization in deep learning. In *International Conference on Learning Representations*.
>
> [C2] Zhang, C., Bengio, S., Hardt, M., Recht, B. and Vinyals, O. (2017). Understanding deep learning requires rethinking generalization. *International Conference on Learning Representations*.
>
> 3. Typos and Remarks.
>
> Thank you for your thorough feedback. We will diligently address each of the typos you've highlighted. Furthermore, [A] extensively covered implicit regularization within linear models and delved into L1 penalty and early stopping. This reference is incredibly valuable to us, and we will certainly include it in the revised version. We appreciate your keen observation and for bringing this to our attention.

---

> > ### Comment · Reviewer_Z81p · 2023-08-21
> > **Acknowledging rebuttal**
> >
> > I thank the authors for their feedback. Other reviewers have raised valid critics, which I believe have been addressed by the authors.

---

> > > ### Author Response · Authors · 2023-08-21
> > > **Reply to the Comment**
> > >
> > > We greatly appreciate your feedback, which has provided us with a deeper understanding and has guided us in considering future directions for our work.

---

### Official Review · Reviewer_i2nR · 2023-07-25

**Soundness:** 3 good
**Presentation:** 2 fair
**Contribution:** 3 good
**Rating:** 6
**Confidence:** 4

**Summary:**

This paper studies the implicit regularization in over-parameterized (sparse) support vector machine. The paper by nature is an extension of Vaskevicius et al [28], applying the quadratic reparametrization on SVM (hinge loss). Due to the non-differentiability of hinge loss, Nesterov’s smoothing is applied to enhance the computational efficiency. The authors then propose a regularization free gradient descent algorithm and show that the sparse solution is obtained with early stopping. The convergence result is demonstrated via numerical simulations. This is an active area of research, and the contribution is relevant. My concern is more about the setting/significance of this work. The authors claim that “this is the first study that investigates implicit regularization via gradient descent and establishes the near-oracle rate specifically in classification” (line 49). First of all, the implicit regularization via gradient descent in classification has been studied in [12, 27] for logistic/exponential losses. Therefore, the contribution to hinge-loss is appreciated (with the remedy of Nesterov's smoothing). Secondly, when it comes to classification problems, sparsity is relatively minor and the implicit regularization effect is shown via max-margin/separability. More discussion would be helpful about the sparse SVM.

These two papers [W]&[Z] below may be closely related to the implicit regularization effect studied in this paper but missed in the related works.

[W] B. Woodworth et al, Kernel and Rich Regimes in Overparameterized Models, COLT, 2020.

[Z] P. Zhao et al, High-Dimensional Linear Regression via Implicit Regularization, Biometrika, 2022.


**Strengths:**

1. The implicit regularization under hinge-loss is novel.
2. Some theoretical results in this paper are interesting.


**Weaknesses:**

1. The importance of sparsity in SVM and how that affects the classification performance is not discussed.
2. Some metrics used in simulations are not defined.

**Questions:**

1. I didn’t find the definition of either $\beta^\star$ or the data generating mechanism. This may cause a big concern if the authors implicitly assume $y=X\beta^\star$ somewhere, which would be very restrictive. In line 258, the simulation follows a logistic model. Is that also the case for theorem 2? If $\beta^\star$ is a minimizer of the population version of the hinge loss, is that for Eq. (1) or Eq. (3)?
2. Formatting issue in [11]
3. Is there a reason that the oracle method performs the worst in Figure 2 (b)?
4. Is Nesterov’s smoothing necessary here? How about plain sub-gradient descent with quadratic parameterization?
5. For Proposition 2, isn’t the upper bound of t in Proposition 1 still needed? I didn’t check the proof details, but intuitively the convergence on the support is only guaranteed while the error accumulation outside the support remains small. Correct me if I’m wrong.

**Limitations:**

The significance/importance/potential application of this work is not well discussed.

---

> ### Author Rebuttal · Authors · 2023-08-09
>
> Thank you for your thoughtful feedback on our paper. We'll carefully review each point and provide explanations. If you have any new questions or ideas, please don't hesitate to share them with us.
>
> 1. More discussion of the sparse SVM.
>
> In modern applications, we frequently encounter classification challenges amidst an abundance of redundant features. The prevalence of sparse-scale data is particularly pronounced in fields like finance, document classification, image analysis, and gene expression analysis. To illustrate, in genomics, only a handful of genes from thousands of candidates are utilized to construct a classifier for disease diagnosis and drug discovery. Similarly, in spam classification, an accurate classifier is sought using a relatively small selection of words from a dictionary containing numerous different terms. In such scenarios, there are potential limitations associated with standard SVMs or SVMs augmented with explicit regularization (such as L2 norm). From an applied perspective, incorporating sparsity into SVMs presents an intriguing avenue for exploration.
>
> Conversely, it's well-established that sparsity considerations have undergone extensive and in-depth exploration in regression in recent years. However, the exploration of theoretical assurances for sparse SVMs is comparably less extensive. Typically, the focus revolves around generalization error and empirical risk analysis, with even fewer discussions on variable selection and error bound. Our study of implicit regularization in classification problems, as presented in this paper, stands as a complementary endeavor to the theoretical pursuits within the realm of sparse SVMs.
>
> 2. Differences with existing studies of implicit regularization in classification.
>
> Compared to these existing work [12,27], our approach adds a crucial extra step – quadratic parameterization. This step involves using two vectors, w and v, and combining them in a special way: $\beta=w\odot w-v\odot v$. Although this unique step makes our theory more intricate, it lets us precisely understand how gradient descent changes over time, which is explained more in Appendix B. we can theoretically determine How the real signal changes at certain stages and how the error is controlled, which helps us to understand more clearly how the parameters change in gradient descent, which is difficult to see in previous classification work.
>
> Additionally, while previous work often relates the speed of convergence to the number of steps taken, like ${\cal O}(1/\log t)$, our paper's theory showcases that the gradient descent method can achieve a rate of convergence that doesn't depend on the step count $t$, within a certain range. This is a theoretical improvement in how we understand the process.
>
> 3. Supplement to the real coefficients $\beta^*$.
>
> Thank you very much for pointing this out for us, we omitted the definition of  the true parameter $\beta^*$ in the main text.The key result of the paper is an error bound $||\beta_t-\beta^*||^2$, where $\beta^*$ is the minimizer of the population version of the hinge loss function (with respect to $\beta$, without $\ell_1$ norm), that is, $\beta^*=\arg\min_{\beta}{\mathbb E}(1-yx^T\beta)_+$. We will certainly make the necessary revisions in the updated version. Moreover, our data generating mechanism in the simulation is within the domain of Theorem 2.
>
> 4. Reason why the oracle method performs the worst in Figure 2(b)
>
> In high-dimensional settings, both the GD estimator and the lasso estimator tend to overestimate the number of signals, which is more than the true number of signals. Consequently, both estimators incorporate a larger number of features, potentially resulting in an improved but ultimately misleading predictive performance.
>
> 5. Necessity of Nesterov’s smoothing
>
> In over-parameterized applications, like the case of gene data classification, the non-smooth nature of the hinge loss poses significant computational and accuracy challenges. Standard first-order methods like the subgradient and stochastic gradient techniques don't achieve rapid convergence for large-scale problems. Second-order methods like Newton's approach and simulated Newton's method can be used, but they're computationally demanding due to the need for the Hessian matrix in each iteration. As a result, a common approach to tackle these complexities in large-scale problems is to 'smooth' the hinge loss. Numerous smoothing methods have been proposed and proven effective in real-world data applications. Given this landscape, using Nesterov's smoothing makes sense in over-parameterized scenarios. It's practically essential.
>
> Additionally, incorporating Nesterov's smoothing aids in our theoretical deductions. If we were to directly analyze the gradient algorithm with quadratic parameterization for the non-smooth hinge loss (which is computationally possible), its non-differentiability would complicate the theoretical deductions. In essence, Nesterov's smoothing plays a significant role both computationally and theoretically.
>
> 6. Explanation of Proposition 2
>
> Your observation is spot-on, and we appreciate you bringing it up. There is indeed an upper limit on the number of iterations to manage the size of the error term outside the support set. The confusion might arise because we presented the error term and strong signal term in two separate propositions. This layout could give the impression that Proposition 2 lacks an upper bound on the number of iterations. We will address this issue and rectify both propositions in the revised version. Thank you for highlighting this, as it helps us enhance the clarity of our work.
>
> 7. References and Formatting issue
>
> We take note that you have provided us with new literature [W], related to implicit regularization, which we will cite in the revised version. Meanwhile, we will revise the formatting issue you mentioned, thanks for pointing it out!

---

> > ### Comment · Reviewer_i2nR · 2023-08-15
> >
> > My major concern about sparse SVM was resolved by the authors' reply to AC's follow-up questions. I'd like to see those discussions added to the main context, which would help highlight the importance of this work.
> >
> > The definition of $\beta^\star$ and an accurate statement of proposition 2 are crucial, please be sure to revise them.
> >
> > After all, I will increase my score to 6, as the authors addressed most of my concerns.

---

> > > ### Author Response · Authors · 2023-08-16
> > > **Reply to the Comment**
> > >
> > > Thank you very much for your reply, we will add these discussions and the relevant definitions in the revision.

---

### Author Rebuttal · Authors · 2023-08-09

We highly appreciate the invaluable feedback we received from reviewers for our work. Your comments aid us in identifying areas where we can enhance our research and make our findings clearer and more accessible to our readers. We have considered all the comments and summarized the responses below. We have selected five key issues to provide a global explanation here.

1. **Discussion of Sparse SVM.**

Several reviewers have highlighted the significance of investigating sparse SVM, prompting us to delve deeper into this topic. In modern applications, the challenge of classification arises when dealing with an abundance of redundant features. such as in the fields of face detection, text classification, image classification, bioinformatics, handwriting recognition, and medical data analysis. To illustrate, in medical imaging, our aim is to construct a classifier using specific pixels from high-dimensional data, while in text analytics, we seek to develop an accurate classifier using a modest subset of words from a vast dictionary (for instance, the libsvm website's news20 dataset comprises 19,996 samples and a feature count as high as 1,355,191, with a sparsity of just 0.034%). Acknowledging sparsity within SVMs adds an intriguing dimension to the study of practical applications.

Conversely, although sparsity in regression has garnered extensive attention in recent years, the corresponding theoretical exploration in classification problems remains relatively limited. The focus generally gravitates towards analyzing aspects like generalization error rate and empirical risk, with less emphasis on error bound. Thus, there exists substantial room for theoretical advancement in the realm of sparse SVM.

In summary, our study of implicit regularization in classification problems in this paper is a very meaningful topic, both as a complement to the theoretical work on sparse SVM and as a new algorithm for the real-world problem of large-scale data classification.

2. **Novelty of our work.**

Our work is the first to design unregularized gradient-based algorithm for SVM by leveraging over-parameterization and provides relevant theoretical guarantees for implicit regularization.

The reparameterization technique has not been applied to classification problems, although it has been widely used in the study of regression. Reparametrization is not computationally burdensome, and although it creates a snag in the theoretical derivation, with reparametrization, we can theoretically analyse how strong signals, weak signals, and error terms change during the iteration process, and we can clearly study the dynamics of gradient descent to get a clearer picture of the algorithmic iteration process, which has never been done in previous papers. With the help of reparameterization, we introduce an error bound for the iterative solution that is independent of the number of iterations $t$.

In addition, we add Nesterov's smoothing, which is a very clever step that computationally overcomes the non-differentiability of the hinge loss and facilitates our theoretical proofs.

3. **Further discussion of assumptions**.

Several reviewers have asked about assumptions, and it is a good opportunity for us to make more explanations of assumptions. Firstly, the assumptions we made in the theory section were mainly motivated by theoretical proofs, such as that the initial size $\alpha$ controls the size of the error term, and that we need to make assumptions about it. However, our assumptions are not strict, for example, ours for $\alpha$ is $\alpha\lesssim 1/p$.

In our experiments, we have found that the theoretical constraints can be relaxed, for example, we can still get good estimation results with heavy-tailed distributions, which means that our method will not be constrained by the assumptions in practical applications.

4. **Addressing unclear and inappropriate expressions in the text.**

We acknowledge the presence of unclear and inappropriate expressions, as well as writing errors in the text. We are committed to rectifying each of these issues in the revised version to enhance the overall quality of the article. We extend our gratitude to the reviewers for their diligent review, as their feedback will undoubtedly contribute to the article's improvement.

5. **Future directions.**

Firstly, as we mentioned above, the assumptions are mainly derived for theoretical considerations and can be relaxed in practical applications, and we would like to explore to what extent these assumptions can be relaxed, which is one of the future work mentioned in other studies on implicit regularization;

Secondly, as suggested by the reviewers, we can consider extending the current study to nonlinear SVM. This could involve incorporating kernel technique to delve into the realm of implicit regularization in nonlinear classification.

(A response to Reviewer 6PiA is included in the pdf file we have provided.)

---

> ### Comment · Area_Chair_TonW · 2023-08-13
> **Follow-up questions**
>
> Dear authors,
>
> I thank you very much for your detailed response to the concerns of reviewers.
> As AC, I would like to ask follow-up questions to have a better understanding of contributions.
>
> 1. Implicit bias is often formulated for widely used algorithms to justify their statistical power helping with generalization. Yet, the design of algorithms with an implicit bias is not motivating unless (i) they are used in practice, or (ii) they outperform algorithms with an explicit regularization or have a better computational or statistical property.  What is the advantage of your algorithm compared to regularization?
>
> 2. Dual formulation of SVM contains an $\ell_1$ regularization term which imposes sparsity. In that regard, there are a few support vectors even for considerably large datasets. Considering this sparsity, what is the advantage of sparse SVM studied here? I checked the cited papers but I could not find a solid motivation.

---

> > ### Author Response · Authors · 2023-08-15
> > **Reply to Follow-up Questions**
> >
> > We appreciate your comments, which have deepened our understanding of our paper. We answer these two questions separately in the hope of providing you with a clearer insight into our contribution. If you have any additional questions or ideas, please don't hesitate to inform us.
> >
> > 1. Motivated by the extensive research on implicit regularization [1,2,3,4,5] and the intensive study of SVM in practice, we leverage re-parametrization in SVM to design regularization-free gradient descent algorithm. Our approach goes beyond simply extending related work by addressing challenges posed by non-differentiability of the hinge loss and complex gradient descent dynamics in classification through smoothing methods and detailed derivations.
> >
> >    The main advantage of our algorithm is that it significantly outperforms algorithms with explicit regularization in terms of applications. We illustrate this through three aspects: first, in terms of estimation error, numerical studies show that our method  generalizes better than $\ell_1$-regularized method. Second,  in terms of variable selection, numerical results also show that our method can reduce false positive rates greatly. Third, since we only need to run gradient descent and the gradient information can be efficiently transferred among different machines, our method is easier to be paralleled and generalized to large-scale applications. This makes our method widely applicable in modern machine learning applications. Similar observations have also been made under many other learning tasks [1,3,4].
> >
> >    From the theoretical perspective, under mild conditions, our estimator enjoys near-oracle statistical rate whereas the most commonly used $\ell_1$-regularized method always results in large bias. This advantage has also been particularly emphasized in related studies [3,4].
> >
> > [1] Vaskevicius, T., Kanade, V., & Rebeschini, P. (2019). Implicit regularization for optimal sparse recovery. *Advances in Neural Information Processing Systems*, *32*.
> >
> > [2] Li, J., Nguyen, T., Hegde, C., & Wong, K. W. (2021). Implicit sparse regularization: The impact of depth and early stopping. *Advances in Neural Information Processing Systems*, *34*, 28298-28309.
> >
> > [3] Fan, J., Yang, Z., & Yu, M. (2022). Understanding implicit regularization in over-parameterized single index model. *Journal of the American Statistical Association*, 1-14.
> >
> > [4] Zhao, P., Yang, Y., & He, Q. C. (2022). High-dimensional linear regression via implicit regularization. *Biometrika*, *109*(4), 1033-1046.
> >
> > [5] Li, J., Nguyen, T. V., Hegde, C., & Wong, R. K. (2023). Implicit regularization for group sparsity. *International Conference on Learning Representations*.
> >
> >
> >
> > 2. By solving the dual optimization of the classical SVM, it is known that learned estimator can be expressed in terms of only a subset of the training points  from the Karush–Kuhn–Tucker (KKT) condition, which are known as the "support vectors''. We want to clarify that this phenomena of  support vectors illustrates the sparsity of the contribution of  the sample points, and represents the sparsity with respect to the sample size $n$. This totally differs from the sparse SVM problem considered in our paper,  which focuses on the sparsity with respect to the collected features (variables) $p$. This paper is motivated by   the high dimensional data   that  in many practical applications,  the challenge of classification is   the presence of a very large number of redundant variables (features), which may adversely affect the classification performance. For example, a gene data set typically contains more than $10000$ genes, and it is crucial to build a classifier using a few number of genes from the collected genes for the purpose of disease diagnosis. Thus, the sparse SVM problem considered in our paper is of  fundamental importance and has attracted tremendous interest in literature [6,7,8,9] due to its power and popularity. We will highlight the difference between the considered sparse SVM problem and the classical one  and provide more detailed clarification on the motivations of the considered problem in the revised version.
> >
> > [6] Zhang, X., Wu, Y., Wang, L., & Li, R. (2016). Variable selection for support vector machines in moderately high dimensions. *Journal of the Royal Statistical Society Series B: Statistical Methodology*, *78*(1), 53-76.
> >
> > [7] Peng, B., Wang, L., & Wu, Y. (2016). An error bound for l1-norm support vector machine coefficients in ultra-high dimension. *The Journal of Machine Learning Research*, *17*(1), 8279-8304.
> >
> > [8] Lian, H., & Fan, Z. (2018). Divide-and-conquer for debiased l1-norm support vector machine in ultra-high dimensions. *Journal of Machine Learning Research*, *18*(182), 1-26.
> >
> > [9] Dedieu, A., Mazumder, R., & Wang, H. (2022). Solving L1-regularized SVMs and related linear programs: Revisiting the effectiveness of Column and Constraint Generation. *The Journal of Machine Learning Research*, *23*(1), 7389-7429.

---

### Decision · Program_Chairs · 2023-09-21

**Decision:**

Accept (poster)

**Comment:**

The proposal is a novel regularization-free algorithm for sparse SVM. Although there were concerns about the explicit regularization and applications of sparse SVM,  the rebuttal has convinced reviewers that contributions are sufficient in the line of implicit regularized learning. The paper may benefit from (i) more motivation on regularization free algorithm design (ii) discussions on the Frank-Wolfe method which also has a bias to sparse solutions.